# Frugal Prompting for Dialog Models

**Bishal Santra**[1*], **Sakya Basak**[2*], **Abhinandan De**[1], **Manish Gupta**[2], **Pawan Goyal**[1]
[1]IIT Kharagpur, [2]Microsoft
{bishal.santra@,abhinandan0316@,pawang@cse}.iitkgp.ac.in
{sakya.basak,gmanish}@microsoft.com

## Abstract

The use of large language models (LLMs) in natural language processing (NLP) tasks is rapidly increasing, leading to changes in how researchers approach problems in the field. To fully utilize these models' abilities, a better understanding of their behavior for different input protocols is required. With LLMs, users can directly interact with the models through a text-based interface to define and solve various tasks. Hence, understanding the conversational abilities of these LLMs, which may not have been specifically trained for dialog modeling, is also important. This study examines different approaches for building dialog systems using LLMs by considering various aspects of the prompt. As part of prompt tuning, we experiment with various ways of providing instructions, exemplars, current query and additional context. The research also analyzes the representations of dialog history that have the optimal usable-information density. Based on the findings, the paper suggests more compact ways of providing dialog history information while ensuring good performance and reducing model's inference-API costs. The research contributes to a better understanding of how LLMs can be effectively used for building interactive systems.[1]

## 1 Introduction

Large Language Models (LLMs) have rapidly transformed the landscape of natural language processing (NLP) research through emergent capabilities like prompt-based learning, in-context learning (ICL), and conversational capabilities (Wei et al., 2022). While these novel approaches are being applied to various domains and tasks with an unrealistic speed and effectiveness, many dimensions

---

*Equal contribution

[1]We make the code publicly available on following two repos: 1) https://github.com/bsantraigi/Frugal-Prompting 2) https://github.com/bsantraigi/Frugal-Prompting-Analysis.

of LLMs remain unexplored. For instance, since we do not need to follow a fixed schema for the textual inputs anymore (like standard supervised learning for text), the ways in which input-text can be presented, and its impact on task performance is an essential aspect that needs to be investigated. Additionally, as various LLM-inference APIs are becoming available for a price, trade-off between performance gain and prompting (inference) cost is another dimension that requires attention.

While efforts have been made to reduce inferencing costs of Transformer (Vaswani et al., 2017) models, these contributions have mostly been at the architecture level and require access to the model weights and source codes (Tay et al., 2022). As many models like GPT-3 (Brown et al., 2020), CODEX (Chen et al., 2021a), LaMDA (Thoppilan et al., 2022), and PaLM (Chowdhery et al., 2022) are now closed source, it is not possible for the end-user to optimize the model's costs using these approaches.

In recent prompt engineering literature, the focus has been on optimizing the prompt to improve downstream task accuracy (Chung et al., 2022; Wei et al., 2021), with the majority of past efforts targeting single-turn tasks (e.g., classification, reading comprehension, question answering, etc.). However, for longer inputs, another critical factor is the inferencing API cost, which has largely been ignored in prior works. This is especially true for interactive or dialog tasks.

This paper explores the trade-off between cost and performance for LLMs in a prompt-based/in-context learning (ICL) setup. We propose the idea of *frugal prompting* in the context of dialog models, which involves input optimization methods to maintain performance gains while minimizing costs. To compare effectiveness of input representations for in-context learning based methods while considering both cost and task performance, we introduce a new metric called *Usable Information*

*Density* (UID). Using this metric, we gain insights into the capabilities of various ICL model families for understanding and accessing information from different input representations.

Overall, we make the following contributions in this paper. (1) We explore the effectiveness of various ICL models and input formats for dialog modeling. (2) We propose a new metric, UID, that captures the tradeoff between accuracy and length for various (input format, ICL model) combinations. (3) Extensive experiments on two benchmark dialog datasets (MSC and TC) and four ICL models show that (a) Adding more context as part of the input does not necessarily improve UID by similar amounts across all ICL models. (b) For most ICL models, using the most semantically related utterance from dialog history is more cost effective compared to using full history, summarized dialog history or most recent utterance.

## 2 Literature Review

**Large language models (LLMs) for Dialog modeling**: A large number of recent dialog generation models have been based on pretrained LLMs like DialoGPT (Zhang et al., 2019), Plato (Bao et al., 2019), Blenderbot (Roller et al., 2021; Shuster et al., 2022), Meena (Adiwardana et al., 2020), and LaMDA (Thoppilan et al., 2022) which use the transformer architecture. Although large scale pretrained models like 175B Blenderbot-3 (Shuster et al., 2022) or the 137B LaMDA (Thoppilan et al., 2022) lead to high accuracies across multiple dialog datasets, these approaches can be prohibitively expensive due to the ever-increasing size of models. In-context learning (ICL) (Brown et al., 2020) with prompt-based models helps avoid expensive finetuning. Further, better accuracies have been obtained using instruction finetuning in models like T0 (Sanh et al., 2022), FLAN (Chung et al., 2022), Tk-Instruct (Wang et al., 2022), etc. But the increased inference costs due to large prompts sizes remains an open challenge.

**Ways to optimize computation for LLMs**: Following environmental impact discussion of the training process of these LLMs (Strubell et al., 2019), multiple studies have proposed these two main lines of work on optimizing costs of LLMs: (1) Model distillation-based (Hinton et al., 2015; Sanh et al., 2019; Gou et al., 2021; Gupta and Agrawal, 2022) methods train a smaller, simplified model to approximate the predictions of a larger,

more complex model. (2) Efficient transformer architectures (Kitaev et al., 2020; Wang et al., 2020; Zaheer et al., 2020; Beltagy et al., 2020) aim to reduce the quadratic complexity of the standard transformer architecture by using more efficient self-attention mechanisms. In this paper, we examine the costs associated with the use of prompts in LLMs and suggest a new method for assessing the cost-performance trade-offs involved, as well as strategies for optimizing the inference cost with regard to the inputs. Please refer to Appendix F for more detailed literature review.

## 3 Prompting Methods for Dialog Systems

We first present the necessary ingredients of a prompt for dialog systems. Next, we discuss recipes for manual and algorithmically optimized prompts. Lastly, we present ways of effectively including context information as part of prompts.

### 3.1 Prompt Ingredients for Dialog Systems

To build a prompt-based dialog system using LLMs, the following components or information sources are an important part of the prompt template.
(1) **Task Instruction:** The instruction is used to explain the task of a dialog response generation model. We also assign a system-role for the LLM (also called Person2) to play through the instruction for example the role of "an automated chat system".
(2) **Dialog Context:** As part of the dialog context, several components can be included like dialog history, persona information and Person1's latest utterance.
(a) **Dialog history**: This refers to the past conversation between Person1 and Person2 that provides the context for the current conversation.
(b) **Background Information (BI)**: We also make use of some additional information like persona or knowledge sections when available. **Persona** is a fictional representation of a user consisting of series of sentences describing their characteristics, events and opinions. This is used to create a personalized experience during a conversation. **Knowledge sections** are short paragraphs from different data sources (Wikipedia, Reddit, and Washington Post) that are related to the topic of the conversation. We experiment with various combinations of different pieces of information to understand their impact on the accuracy versus inference cost.
(3) **Person1's latest utterance**: This is the most recent statement or question uttered by Person1 in

| Template | Prompt |
|---|---|
| **Here is a summary of the conversation between Person1 and Person2:** [S] | **Here is a summary of the conversation between Person1 and Person2:** Person1 wants to go back to college to learn more about accounting. Person2 wants to study education so Person2 could teach art. Person1 thinks it's never too late for a career change. |
| **Based on the dialog between the Person1 and the Person2 so far, try to anticipate what the Person2's response might be to the Person1's next statement.**
**Person1:** [U]

**Person2:** | **Based on the dialog between the Person1 and the Person2 so far, try to anticipate what the Person2's response might be to the Person1's next statement.**

**Person1:** "I've been here five years. FIVE long years. It's not the most rewarding job but it's steady and reliable so I never really looked for anything else, but I'm starting to want a change."
**Person2:** |
| **Generation** | |
| [R] | I think you should look into a career change. It's never too late to learn something new. |

Table 1: Prompt template and an instantiation with summary of dialog history as dialog context and without exemplars or background information. This is perplexity optimized using FLAN-T5-XL model. More examples are provided in Appendix G.

a dialog, that prompts the Person2's response.

(4) **Exemplars:** Although most recent LLMs are capable of solving tasks just using instructions (due to RLHF and instruction-finetuning), providing examples along with task description may help improve performance. We test our prompt-based models in two configurations with respect to number of examples: *zero-shot* and *few-shot*.

An example prompt is shown in Table 1. A full list of all the prompts used in our experiments can be found in the Appendix G.

| |
|---|
| **Automated Chat System:** |
| **Learn from the below example on how to generate consistent and diverse responses between Person1 and Person2 given background details along with summary. Example:** |
| **Here are some background details about Person1:** $[BI(P_1)_E]$
**Here are some background details about Person2:** $[BI(P_2)_E]$ |
| **This is a summary of a dialog exchange between Person1 and Person2:** $[S_E]$ |
| **Given the background details and the summary of the dialog exchange between Person1 and Person2, give a consistent and diverse response to the following dialog by Person1.** |
| **Person1:** $[U_E]$
**Person2:** $[R_E]$ |
| **Now try it yourself:** |
| **Here are some background details about Person1:** $[BI(P_1)]$
**Here are some background details about Person2:** $[BI(P_2)]$ |
| **This is a summary of a dialog exchange between Person1 and Person2:** $[S]$ |
| **Given the summary of the dialog exchange between Person2 and their background details, give a consistent and diverse response to the following dialog spoken by Person1.** |
| **Person1:** $[U]$
**Person2:** |

Table 2: Manually engineered prompt template with summary of dialog history, persona and latest person1 utterance as dialog context and with one exemplar. $[BI(P_1)]$ is person1's persona, $[BI(P_2)]$ is person2's persona, $[S]$ is summary of the dialog history, $[U]$ is the latest person1 utterance, $[R]$ is the person2 response. $\cdot_E$ implies corresponding elements are for the exemplar.

### 3.2 Manual versus Perplexity Prompts

We experimented with two ways to design prompt templates: manual and automatically optimized prompts using a perplexity-based search method.
**Manually Designed Prompts**: Manual prompts were designed keeping in mind general principles of prompt design (Liu et al., 2023) like role based prompting (Schulhoff and Contributors, 2022), specifically adding requirements like "generate a consistent, diverse response" so as not to get repetitive, dull responses and maintain consistency with respect to the current utterance and context. Table 2 illustrates one of our manually designed prompt template, with summary of dialog history, persona and current user utterance as dialog context and with one exemplar.

**Perplexity Optimized Prompts**: We followed the strategy highlighted in Gonen et al. (2022) which claims that the performance of a prompt is coupled with the extent to which the model is familiar with its language, and this can be measured by the perplexity of the prompt. Given an LLM, we took the manually engineered prompt template, and created candidate prompt variants by using GPT3 and back translation. Further, we instantiated all such prompt templates using 100 instances (with full prompt sequence, including the input itself, and without the label), and computed average perplexity per template using the LLM. The lowest perplexity template was chosen.

### 3.3 Optimizing the Dialog History Input

**Redundancies in conversations:** In conversational agents, dialog history plays a crucial role in generating meaningful responses. It provides context and continuity, and enables the agent to remember previous interactions with the user. However, the dialog history can also be redundant, especially when it contains back-channeling, clarification, and mistake correction. While these elements are necessary for a natural and useful conversation, they increase the length of the dialog history without adding any new information. In addition, responses from some dialog models (like Instruct-

GPT (Ouyang et al., 2022)-based models – text-davinci-003) could be elaborate and long.

**Shortening Dialog Histories:** To reduce the prompt length, we can compress the dialog history by removing redundancies. The goal is to give the agent only the parts that are relevant and informative for generating the next response. Two possible approaches to compress the dialog history into a shorter and more informative representation are selection and summarization.

- Selection: Two possible ways to select parts of dialog history are as follows. (1) **Recent-$k$**: The simplest approach is to use a fixed-length dialog history from the most recent utterances. However, this approach may not be optimal, as users may refer back to context beyond the fixed length window and expect the system to understand. (2) **Semantic-$k$**: In this approach, the most relevant $k$ utterances from the dialog history are selected with respect to the current utterance. This method is simple, but its performance depends on the quality of the similarity measure used. We used the average of the similarity obtained using SimCSE model (Gao et al., 2021) and Sentence Transformers (Reimers and Gurevych, 2019) to measure the overall similarity between utterances.

- Summarization: An alternative approach is to use a summary of the full dialog history. We considered two Transformer-based encoder-decoder abstractive summarization models (BART (Lewis et al., 2019) and Pegasus (Zhang et al., 2020)) finetuned on generic as well as dialog datasets like CNN/DailyMail (Hermann et al., 2015), SAMSum (Gliwa et al., 2019) and Dialog-Sum (Chen et al., 2021b). These methods are more complex, but they can generate a summary that is more informative and short.

**Shortening Background Information:** Often dialog datasets also include other background information like persona information (Xu et al., 2022), reading sets (Gopalakrishnan et al., 2019) and knowledge facts (Dinan et al., 2018). Transformer-based encoder-decoder abstractive summarization models (BART (Lewis et al., 2019) and Pegasus (Zhang et al., 2020)) can be used to shorten such background information as well.

## 4 Experimental Setup

### 4.1 Datasets

We experiment with two dialog datasets for comparing various methods on accuracy versus inference cost for prompt-based dialog systems: Multi-session Chat (MSC) (Xu et al., 2022) and Topical Chat (TC) (Gopalakrishnan et al., 2019). We chose these datasets because of their varying characteristics and the length of the dialog history.

The MSC dataset consists of multiple chat sessions whereby the speaking partners learn about each other's interests and discuss the things they have learnt from past sessions. Each user participating in these sessions (or conversations) is asked to play a role (persona) while having the conversation. On the other hand, in the TC dataset, each pair of users is assigned one or more topics along with some facts or knowledge about the topic, and the users are asked to have a conversation about the topic. Users have no persona in the TC dataset but there are knowledge sections associated with the conversations. The test set contains 16,299 and 7,512 context response pairs in the MSC and TC datasets, respectively. Also, there are 11.9 and 20.0 average number of utterances in full conversations in the MSC and TC datasets, respectively.

Since we do not train or finetune any specific models, we do not use train splits of these datasets. For perplexity-based prompt optimization, we use validation splits of these datasets. We discuss detailed preprocessing steps in the Appendix A.

### 4.2 Summarization of Dialog and Background Information

We used BART and Pegasus models for summarization. However, in dialog summarization, the objective is to distill the most important information or key points from a conversation, which can be quite challenging because conversations tend to be more dynamic and context-dependent than normal documents. Unlike traditional summarization, in dialog summarization, there is a greater emphasis on preserving the coherence and context of the conversation. Hence, we used dialog summary datasets like DialogSum, SAMSum and CNN/DailyMail to finetune abstractive summary models like Pegasus and BART and picked up the best model for use in terms of summarization performance by calculating ROUGE metric on dialog summarization data.

We process the DialogSum and SAMSum datsets to remove all conversation instances having more than two speakers and normalized the speaker names to Person1 and Person2 so that the model does not hallucinate random names during summary generation.

Overall, we train three models: (1) BART-D: facebook/bart-large model finetuned on Dialog-Sum, with 12 encoder and 12 decoder layers. (2) Pegasus-CD: google/pegasus-cnn_dailymail model (which has been finetuned on CNN-DailyMail corpus, with 16 encoder and 16 decoder layers. (3) Pegasus-DS: google/pegasus-cnn_dailymail model further finetuned on both DialogSum and SAM-Sum data, with 16 encoder and 16 decoder layers. Training hyper-parameters are in Appendix B.

### 4.3 Models and Prompt Design

For this study, we used GPT-3 (text-davinci-003), one of the most prominent models for prompt-based or ICL. Along with GPT-3, we also included other open-source models that are capable of ICL: FLAN-T5 (google/flan-t5-xl), T0 (bigscience/T0_3B), and Tk-Instruct (allenai/tk-instruct-3b-def for zero shot and allenai/tk-instruct-3b-def-pos for few shot). These open-source models are generally smaller in size compared to GPT-3 (175B) and have the capability of ICL through instruction-finetuning based training.

We experiment with several input prompt settings: (1) Zero shot versus few shot. (2) Manually designed versus perplexity optimized prompts. (3) Settings based on usage of dialog history: (a) full history, (b) summarized dialog history (using any of the three summarization models), or (c) Recent-$k$ or semantic-$k$ selection from history. (4) With and without summarized background-information.

In case of few shot, we use only one exemplar since (1) previous work (Madotto et al., 2021) has shown that one exemplar is enough, and (2) we wish to find methods which retain good accuracy with short input lengths. The exemplar is also formatted in the same way as the actual input. For example, if the actual input setting is to use persona with few shot, the exemplar also includes persona information. Similarly, if the actual input setting is to use summarized dialog history with input, the exemplar also includes summarized dialog history.

The exemplar is chosen based on the immediately previous utterances if available, else it is randomly chosen from the dataset. Thus, for each instance, the exemplar is different. For example, consider the Recent-4 few shot setting. Let *ABCDEFG* be the utterances in the conversation. Thus, the instance will have *G* as the target response, and input contains *F* as the current utterance and *BCDE* as the recent-4 dialog history. The input for this instance will also consist of an exemplar where the target response will be *F* and input for exemplar will contain *E* as current utterance and *ABCD* as the recent-4 dialog history.

### 4.4 Metrics

**Performance**  We evaluate the performance of the models using several popular metrics: ME-TEOR, BLEURT and DEB. METEOR (Banerjee and Lavie, 2005) is widely used for various text-generation tasks (machine translation, dialog generation, etc.). It measures lexical-overlap between n-grams of the predicted and ground truth response. BLEURT (Sellam et al., 2020) uses a pre-trained BERT model for evaluating text generation models. DEB (Sai et al., 2020) is a BERT-based dialog metric further pre-trained on dialog data for next response classification using the next-sentence-prediction (NSP) loss.

**Inference Cost**  To evaluate the effectiveness of different prompting methods for dialog systems, we need a metric that takes into account both the performance gain and the inference cost reduction. The cost is measured in terms of the length of the overall input, as longer inputs incur more inference-API costs and also slow down the inference.

We propose a new benefit-cost based metric to simultaneously consider both model performance and the inference cost incurred: the usable-information-density (UID). UID with respect to metric $M$ is defined as $\text{UID}_M(a) = (M_H)^a/L_H$ where $M_H$ is the average performance of the model as per metric $M$, $L_H$ is the overall combined size of input and output averaged across all the test examples, $a$ is a metric-importance parameter. In the main paper, we present results using $a$=1, but show impact of varying $a$ in the Appendix. With $a$=1, UID is defined as the ratio of performance to cost measured in terms of size of the input and output. The UID captures the amount of information, per token, usable by a model (Ethayarajh et al., 2022) for a given input/prompt configuration. The UID metric can be used to evaluate the effectiveness of different prompting methods.

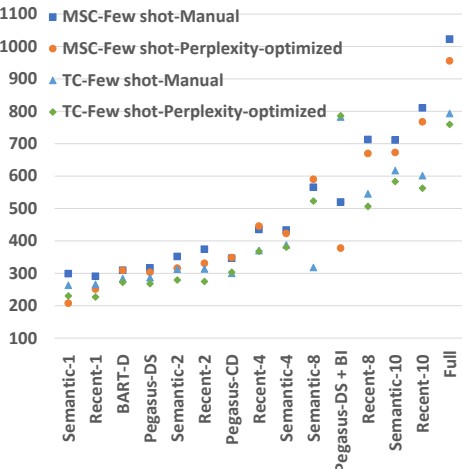

Figure 1: Comparison of average input length for various representations of dialog prompts across the two datasets for the few shot setting. *DH = Dialog History, BI = Background Information.*

## 5 Results and Analysis

### 5.1 Size Comparison across different Input Formats

Fig. 1 shows the variation in the average input prompt size as we vary the prompt constituents, dialog history (DH) and background information (BI), for the few shot. We show a similar figure (Fig. 4) for zero shot cases in the Appendix. We plot the variation for manually engineered as well as perplexity optimized prompts for both the datasets (MSC and TC). $Y$-axis indicates the overall length of the input prompt, which is fed to the large language models (LLM) without further processing. We observe that the complete dialog history is significantly longer compared to summarized or selection forms. Since we use one demonstration exemplar in few shot cases, the few shot prompts are typically twice as long as their corresponding zero shot prompts. Perplexity optimized prompts are slightly shorter than manually engineered prompts on average. Pegasus-DS summarized dialog history is almost 3 times shorter; Pegasus-DS summaries are shorter than Pegasus-CD summaries while BART-D summaries are shorter than Pegasus-DS summaries. Sizes of recent-2 (or semantic-2) are similar to the summarized dialog histories in terms of the final length of the input context to the model. However, we expect that the summarized dialog history will have more useful information stored in a compressed form compared to the greedy choice of only the recent-2 or semantic-2 utterances. In case of Pegasus-DS + BI, we use the BI summarized using Pegasus-CD model. Note

| | | BLEURT | DEB | METEOR |
|---|---|---|---|---|
| MSC | FLAN-T5 | 0.357 | 0.765 | 0.139 |
| | T0 | 0.347 | 0.912 | 0.153 |
| | GPT-3 | 0.386 | 0.929 | 0.182 |
| | Tk-Instruct | 0.355 | 0.832 | 0.133 |
| TC | FLAN-T5 | 0.345 | 0.803 | 0.124 |
| | T0 | 0.321 | 0.868 | 0.133 |
| | GPT-3 | 0.342 | 0.885 | 0.147 |
| | Tk-Instruct | 0.338 | 0.852 | 0.119 |

Table 3: Model comparison based on average performance over history, prompt-type and exemplar settings.

that the summary of background information in TC is much larger compared to that in MSC. For example, Pegasus-DS + BI for TC is as large as full dialog history.

### 5.2 Performance Results and Analysis

In Figs. 2 and 3, we analyze the absolute performances of various LLM model-families using prompts based on various input representations for TC and MSC, respectively. We show results for few shot (FS) as well as zero shot (ZS) cases across three popular metrics – BLEURT, DEB and METEOR. We also show results for manually engineered as well as perplexity optimized prompts averaged across various models (FLAN-T5, T0, Tk-Instruct and GPT-3). Since we do not have access to logits from GPT-3 model, we cannot optimize prompts for GPT-3 using perplexity. Detailed model-wise results are in Appendix Figs. 5 to 8.

For each of these combinations, we show results for different input prompt combinations: (1) Full dialog history, (2) Summary of dialog history using BART-D or Pegasus-DS or Pegasus-CD, (3) Pegasus-DS summary of dialog history as well as Pegasus-CD summary of background information (BI), (4) Recent-$k$ selected dialog utterances, and (5) Semantic-$k$ selected dialog utterances, where $k$ is varied as 1, 2, 4, 8, and 10. Note that we did not experiment with full background information since background information is very large in size, especially for the TC dataset.

As shown in Table 3, GPT-3 generally outperforms the other families of LLMs in terms of absolute performance (DEB, BLEURT, and METEOR) and Tk-Instruct performs the worst. Also, generally we observe the best results with full dialog history for TC in most cases for DEB and METEOR. For MSC, even prompts with summarized history seem to do very well although they are much shorter. Averaged across metrics, we observe that Semantic-$k$ performs better than Recent-$k$ for all values of $k$ (1, 2, 4, 8, 10) for both datasets. Further, while Semantic-$k$ reaches peak perfor-

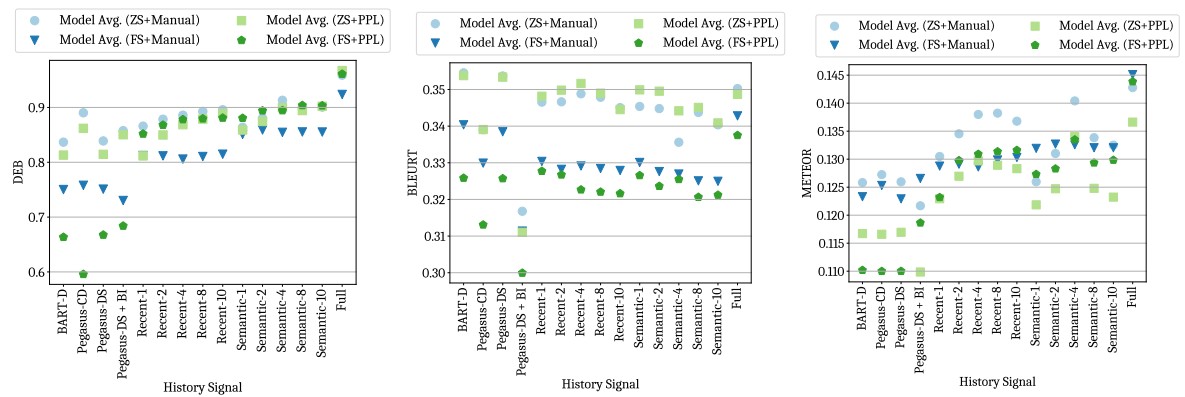

Figure 2: Model averaged performance results for TC Dataset. *DH = Dialog History, BI = Background Information.*

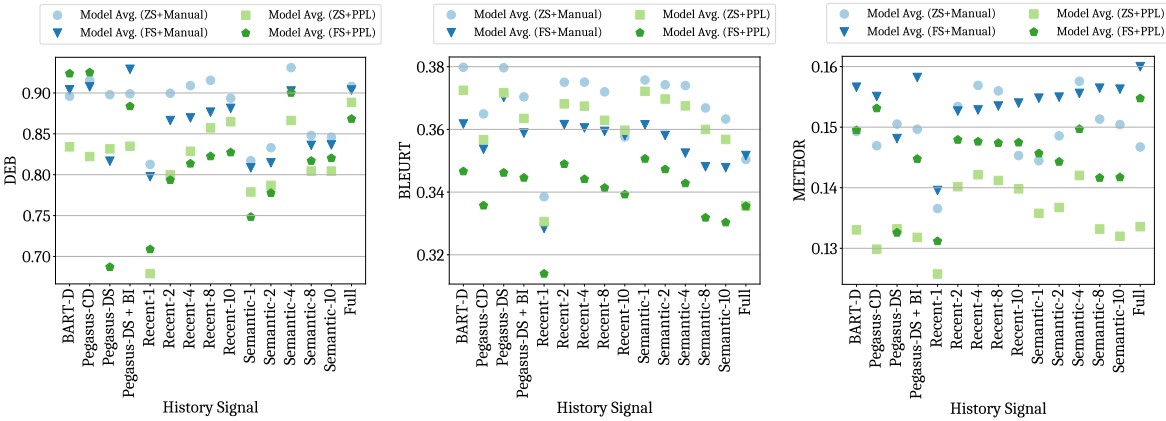

Figure 3: Model averaged performance results for MSC Dataset. *DH = Dialog History, BI = Background Information.*

mance at $k$=4, Recent-$k$ attains the best results at higher values of $k$ (8 or 10). Adding background information (knowledge facts) to Pegasus-DS helps boost DEB and METEOR significantly but hurts BLEURT on average for both the datasets. In MSC, amongst different configurations for history signal, Recent-1 performs the worst on average. In TC, BART-D performs the worst on average. Surprisingly, even though zero shot prompts are almost half the size compared to few shot prompts, zero shot results are better than few shot results, except for perplexity prompts in MSC. Although recent prompt engineering based studies motivate using demonstration examples, it turns out that examples are not very useful for dialog modeling.

Perplexity optimized prompts lead to shorter prompt sizes but not better accuracy values, except for DEB in TC. Since we cannot compute perplexity optimized results for GPT-3, we show results for the remaining three models. We observe that T0 is the best for DEB and METEOR while FLAN-T5 is the best for BLEURT. In both cases, zero shot results are better.

## 5.3 UID Results and Analysis

What is of our main interest is the fact that, in many cases, the summarized dialog-history input is able to attain much of the performance, sometimes even better than the full dialog-history setting, which has a much longer input length. Thus, we are interested in comparing various input representation methods in terms of how much information, per token, a particular LLM model can access, that is converted into better performance on the response generation task. Hence, in this section, we discuss relative importance of different components of the input prompt using the UID as a metric that explains the input prompt size versus the performance tradeoff. We show the UID results averaged across models in Table 4. We also show model-wise results in Tables 5, 6, 7, and 8 for FLAN-T5, T0, Tk-Instruct and GPT-3, respectively. We show results for few shot as well as zero shot cases, and for both the datasets (MSC and TC). We also show UID results across all dialog history, prompt-type and exemplar settings on the three different metrics – BLEURT, DEB and METEOR.

| | History | Manual Prompt | | | | | | Perplexity Prompt | | | | | |
|---|---|---|---|---|---|---|---|---|---|---|---|---|---|
| | | BLEURT | | DEB | | METEOR | | BLEURT | | DEB | | METEOR | |
| | | ZS | FS | ZS | FS | ZS | FS | ZS | FS | ZS | FS | ZS | FS |
| MSC | BART-D | 19.79 | 9.03 | 46.56 | 22.57 | 7.75 | 3.91 | 20.27 | 8.94 | 45.58 | 23.81 | 7.25 | 3.86 |
| | Pegasus-CD | 17.76 | 8.22 | 44.58 | 21.07 | 7.14 | 3.60 | 17.75 | 8.06 | 41.14 | 22.22 | 6.49 | 3.68 |
| | Pegasus-DS | 19.72 | 9.31 | 46.49 | 20.51 | 7.78 | 3.72 | 20.22 | 8.90 | 45.40 | 17.69 | 7.25 | 3.41 |
| | Pegasus-DS + BI | 12.23 | 5.87 | 29.65 | 15.21 | 4.93 | 2.59 | 11.97 | 5.57 | 27.50 | 14.27 | 4.34 | 2.34 |
| | Recent-1 | 22.22 | 9.82 | 53.35 | 23.84 | 8.96 | 4.17 | 22.29 | 9.59 | 46.07 | 21.59 | 8.50 | 3.99 |
| | Recent-2 | 19.38 | 8.72 | 46.49 | 20.88 | 7.92 | 3.68 | 19.65 | 8.52 | 42.88 | 19.33 | 7.49 | 3.60 |
| | Recent-4 | 15.18 | 6.96 | 36.77 | 16.78 | 6.34 | 2.95 | 15.28 | 6.68 | 34.54 | 15.78 | 5.92 | 2.86 |
| | Recent-8 | 11.35 | 5.31 | 27.91 | 12.94 | 4.75 | 2.27 | 11.25 | 5.04 | 26.60 | 12.14 | 4.38 | 2.17 |
| | Recent-10 | 10.05 | 4.85 | 25.08 | 11.92 | 4.08 | 2.08 | 10.14 | 4.59 | 24.40 | 11.19 | 3.94 | 1.99 |
| | Semantic-1 | 23.51 | 10.18 | 51.11 | 22.75 | 9.03 | 4.36 | 24.23 | 10.13 | 50.94 | 21.48 | 8.85 | 4.20 |
| | Semantic-2 | 19.86 | 8.73 | 44.19 | 19.85 | 7.88 | 3.78 | 20.42 | 8.68 | 43.63 | 19.34 | 7.56 | 3.60 |
| | Semantic-4 | 15.50 | 6.98 | 38.57 | 17.87 | 6.52 | 3.08 | 15.62 | 6.87 | 36.93 | 18.02 | 6.05 | 2.99 |
| | Semantic-8 | 11.07 | 5.00 | 25.58 | 12.02 | 4.56 | 2.25 | 11.20 | 4.85 | 25.06 | 11.91 | 4.14 | 2.06 |
| | Semantic-10 | 9.72 | 4.46 | 22.62 | 10.72 | 4.02 | 2.00 | 9.80 | 4.29 | 22.12 | 10.64 | 3.63 | 1.84 |
| | Full | 6.97 | 3.44 | 18.03 | 8.84 | 2.91 | 1.56 | 6.66 | 3.31 | 17.63 | 8.56 | 2.65 | 1.53 |
| TC | BART-D | 24.56 | 11.11 | 57.75 | 24.46 | 8.67 | 4.02 | 25.64 | 11.00 | 59.05 | 22.42 | 8.46 | 3.71 |
| | Pegasus-CD | 18.59 | 8.77 | 48.79 | 20.14 | 6.96 | 3.33 | 19.24 | 8.54 | 48.96 | 16.31 | 6.63 | 3.00 |
| | Pegasus-DS | 24.37 | 11.00 | 57.61 | 24.39 | 8.64 | 3.99 | 25.47 | 10.96 | 58.82 | 22.49 | 8.43 | 3.70 |
| | Pegasus-DS + BI | 7.67 | 3.88 | 20.75 | 9.09 | 2.94 | 1.57 | 7.59 | 3.63 | 20.76 | 8.27 | 2.68 | 1.43 |
| | Recent-1 | 24.17 | 10.62 | 60.36 | 26.08 | 9.09 | 4.13 | 24.98 | 10.83 | 58.37 | 28.06 | 8.83 | 4.05 |
| | Recent-2 | 20.35 | 9.14 | 51.54 | 22.58 | 7.89 | 3.59 | 21.25 | 9.27 | 51.71 | 24.58 | 7.72 | 3.67 |
| | Recent-4 | 15.99 | 7.35 | 40.57 | 17.98 | 6.31 | 2.87 | 16.66 | 7.29 | 41.21 | 19.82 | 6.15 | 2.95 |
| | Recent-8 | 11.92 | 5.54 | 30.56 | 13.65 | 4.73 | 2.19 | 12.24 | 5.47 | 30.85 | 14.93 | 4.52 | 2.23 |
| | Recent-10 | 10.77 | 5.04 | 27.94 | 12.53 | 4.26 | 2.00 | 10.95 | 4.98 | 28.26 | 13.64 | 4.08 | 2.04 |
| | Semantic-1 | 24.02 | 10.50 | 60.06 | 27.05 | 8.76 | 4.19 | 25.48 | 10.73 | 62.63 | 28.87 | 8.88 | 4.17 |
| | Semantic-2 | 20.15 | 8.95 | 51.40 | 23.47 | 7.64 | 3.62 | 21.43 | 9.12 | 53.68 | 25.16 | 7.65 | 3.61 |
| | Semantic-4 | 15.34 | 7.23 | 41.73 | 18.90 | 6.41 | 2.93 | 16.31 | 7.31 | 42.70 | 20.07 | 6.36 | 2.99 |
| | Semantic-8 | 11.57 | 5.29 | 30.20 | 13.92 | 4.49 | 2.15 | 12.01 | 5.30 | 31.14 | 14.94 | 4.34 | 2.14 |
| | Semantic-10 | 10.38 | 4.81 | 27.45 | 12.67 | 4.03 | 1.96 | 10.68 | 4.83 | 28.29 | 13.57 | 3.86 | 1.95 |
| | Full | 8.45 | 4.06 | 23.13 | 10.94 | 3.44 | 1.72 | 8.47 | 4.06 | 23.50 | 11.57 | 3.32 | 1.73 |

Table 4: UID results across four models. Manual prompts are averaged over all 4 models; perplexity optimized prompts are averaged over all models except GPT-3.

Comparing manually engineered prompts versus perplexity optimized prompts, we observe that manually engineered prompts are better on average. We believe this is because perplexity and other metrics (BLEURT, DEB, METEOR) do not show similar correlation with dialog response quality as shown in (Liu et al., 2016).

Across the dialog history types, we make the following observations which hold for both datasets: (1) For most metrics across datasets, we observe that using one semantically related utterance is the best. The UID decreases as we increase $k$. (2) In terms of absolute metrics (Figs. 2 and 3), we observe that Recent-$k$ typically increases with increase in $k$ while Semantic-$k$ peaks at $k=4$ and then drops. But in terms of UID, for both Recent-$k$ and Semantic-$k$, UID reduces with increase in $k$. (3) Adding background information to Pegasus-DS does not help. (4) Amongst summarization methods, Pegasus-DS and BART-D perform better than Pegasus-CD. This is expected since Pegasus-DS and BART-D are both trained on dialog datasets. Using summaries of the dialog history provides better UID results than using the full dialog history. This suggests that models can work more efficiently with summarized input.

As observed from Figs. 2 and 3, few-shot accuracy values are worse than zero-shot, although few-shot are almost twice the size of zero-shot prompts.

This implies that few-shot UID is much smaller than zero-shot UID as can be seen in Table 4.

Overall, we find that using full dialog history, or Semantic-$k$/Recent-$k$ with large $k$ are not very useful from a UID perspective. For both the datasets, it is clear that Semantic-1 and Recent-1 have very good UID values across all models and metrics, with zero-shot being better than few-shot. This suggests that having a smaller but more focused input is recommended for dialog model prompting.

## 6 Conclusion

In conclusion, this paper has explored the trade-off between model performance and cost in interactive tasks where dialog history plays a crucial role. Since recent large language models tend to produce longer dialog responses, using this long dialog history as context for next utterance prediction becomes more expensive. However, the experiments conducted in this study have demonstrated that compressing dialog history can improve model performance without significantly increasing cost. Our findings suggest that the optimal representation of dialog history is one that provides the highest amount of usable information per token. Summaries of dialog history are better than using full history itself. Recent utterance or best semantically similar utterance are both better than summaries. One best semantically similar utterance

is the best from both accuracy as well as usable information perspective. Overall, our results highlight the importance of carefully balancing model performance and cost in interactive tasks that rely on dialog history.

# 7  Acknowledgments

This work was partially supported by Microsoft Academic Partnership Grant (MAPG) 2022. The first author was also supported by Prime Minister's Research Fellowship (PMRF), India.

# 8  Limitations

We experimented with datasets and models trained on languages with limited morphology like English. While we hope that these results will generalize to models trained on multi-lingual datasets; empirical validation needs to be done.

While the study examines TC and MSC, these conclusions may only apply to these datasets and to general open-domain chit-chat dialogue. However, there are many more dialogue settings than just these two. For example, it needs to be validated if the conclusions would apply to more information-critical dialogues (e.g. task-oriented dialogue datasets like MultiWOZ).

For task-oriented dialog systems with well-defined ontologies and belief states, the experimental design would need to be reconsidered, including aspects like prompts, summarization methods, and evaluation metrics. Standard summarization techniques may need to be adapted to better retain key belief state information in the summary. Although we believe that the well-defined ontology could potentially allow further optimization of prompt lengths compared to open-domain dialog. While the lower-level details would differ in applying frugal prompting notions to task-oriented dialogs, we are optimistic that similar beneficial findings around balancing model performance and computational costs could emerge.

# 9  Ethics Statement

In this paper, we studied how to efficiently use dialog generation models. Although we did not explicitly train our own dialog models, we would like to make the readers aware about potential risks in usage of such models. Many pretrained language representation models have learned patterns associated with exposure bias. Interpretability associated with the output is rather limited, hence users should use the outputs carefully. These models generate possible response candidates, and do not filter out any "problematic" candidates. Thus, for applications, where candidate responses could be problematic, (e.g., offensive, hateful, abusive, etc.), users should carefully filter them out before using the output from such models.

All the datasets used in this work are publicly available. We did not collect any new dataset as part of this work.

MSC dataset: The dataset was downloaded from https://parl.ai/projects/msc/. Xu et al. (Xu et al., 2022) describes details about creation of the dataset. Parl.ai makes models and datasets available under MIT License.

TC dataset: The dataset was downloaded from https://github.com/alexa/Topical-Chat. The dataset is available under Community Data License Agreement.

We used 4 models in this work: T0, Tk-Instruct, FLAN T5 and GPT-3 API. T0, Tk-Instruct and FLAN T5 are all provided under Apache 2.0 License on Huggingface. We used the publicly available GPT-3 API by signing up at OpenAI.

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

## A   Data Preprocessing

The MSC dataset is divided into multiple sessions, the first of which uses dialogs from the PersonaChat dataset. Each session has metadata information such as time elapsed from the past conversation and previous dialogs. Examples from Session 1 do not have enough context. Hence, we experiment with examples from sessions 2, 3 and 4 are used, and the results are averaged across the three. As per the dataset construction, a single conversation has been conducted across multiple sessions. Hence, as a first step, we aggregate all

turns for a conversation across sessions 1, 2, 3 and 4 by concatenating them in a temporal way. Further, context-response example pairs for our experiments have been created by considering (i) second utterance of each turn of sessions 2, 3 and 4 as a response and (ii) first utterance of corresponding turn and entire conversation history as context. We also use the persona information as background information when constructing input for various dialog models.

The test split of the TC dataset includes two sections: frequent and rare. This is based on the frequency of the associated entities as observed in the training set. We combine these splits to create our test set and pursue our analysis. The conversations begin with a preprocessed reading set which is retrieved from Wikipedia. Further, context-response example pairs for our experiments have been created by considering (i) second utterance of each turn as a response and (ii) first utterance of corresponding turn and entire conversation history as context.

In both datasets, for each sample, we normalize the utterances by removing trailing whitespaces, and capitalizing first word of every sentence.

## B   Hyper-parameters for training dialog summarization models

We used a batch size of 8 and finetuned the models for 10 epochs. We tried using various learning rates (1e-5, 5e-5, 1e-4, 5e-4, 1e-3) and finally picked a learning rate of 1e-4 since that gave the most optimal performance on the validation set. During training we limited the maximum length of generated summary to 128 and set the number of beams to 5.

## C   Overall input length

Refer to Fig. 4 for a comparison of overall input length for various representations of dialog prompt across the two datasets for the zero shot setting.

## D   Detailed Model-wise Performance Results

In Figs. 5 to 8, we analyze the absolute performances of various LLM model-families using prompts based on various input representations for TC and MSC resp. We show model-wise results for few shot (FS) as well as zero shot (ZS) cases across three popular metrics – BLEURT, DEB and

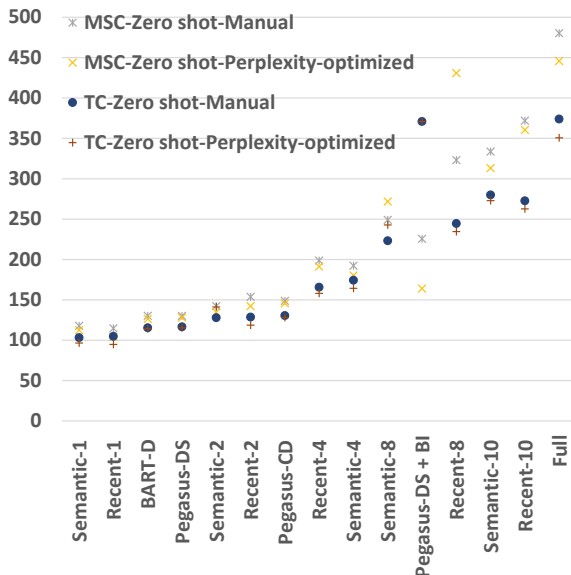

Figure 4: Comparison of average input length for various representations of dialog prompt across the two datasets for the zero shot setting. *DH = Dialog History, BI = Background Information.*

METEOR. We also show results for manually engineered as well as perplexity optimized prompts averaged across various models (FLAN-T5, T0, Tk-Instruct and GPT-3). Since we do not have access to logits from GPT-3 model, we cannot optimize prompts for GPT-3 using perplexity.

## E  Detailed Model-wise UID Results

We show the model-wise UID results in Tables 5, 6, 7, and 8 for FLAN-T5, T0, Tk-Instruct and GPT-3 respectively. We show results for few shot as well as zero shot cases, and for both the datasets (MSC and TC). We also show UID results across three different metrics – BLEURT, DEB and METEOR. For each of these combinations, we show results for different input prompt combinations: (1) Full dialog history, (2) Summary of dialog history using BART-D or Pegasus-DS or Pegasus-CD, (3) Pegasus-DS summary of dialog history as well as Pegasus-CD summary of background information (BI), (4) Recent-$k$ selected dialog utterances, and (5) Semantic-$k$ selected dialog utterances, where $k$ is varied as 1, 2, 4, 8, and 10.

## F  Detailed literature review

### F.1  Dialog modeling

The development of open-domain chatbot systems that possess long-term memory, generate engaging and coherent responses, and perform equally well on a variety of dialog tasks has been a longstanding challenge. Several Seq2Seq models (Serban et al., 2017; Shen et al., 2017; Zhao et al., 2017; Bao et al., 2019; Santra et al., 2021) have been proposed to address the specific properties of dialog modeling. Recently, a significant amount of focus has been on pretraining large dialog generation models like DialoGPT (Zhang et al., 2019), Plato (Bao et al., 2019), Blenderbot (Roller et al., 2021), Meena (Adiwardana et al., 2020), Blenderbot-3 (Shuster et al., 2022) and LaMDA (Thoppilan et al., 2022) using the transformer architecture. Retrieval augmented generation (RAG) has been another prominent approach to tackle the dialog generation task in both large and small-scale models (Wu et al., 2019; Gupta et al., 2020; Cai et al., 2021; Komeili et al., 2021; Zhu et al., 2018). Although large scale pretrained models like 175B Blenderbot-3 (Shuster et al., 2022) or the 137B LaMDA (Thoppilan et al., 2022) lead to high accuracies across multiple dialog datasets, these approaches can be prohibitively expensive due to the ever-increasing size of models. Large model size makes finetuning difficult. Also, in-context learning (ICL) with prompt-based models makes finetuning unnecessary.

### F.2  Prompt-based models

Prompt-based usage of LLMs and in-context learning was introduced by Brown et al. (2020). In prompt-based approach, an LM is adapted to perform a specific task by priming it with instructions and/or examples. Following the success of the in-context learning approach towards generalizing NLP models, various other equally or more capable models based on smaller LMs have also been introduced. Smaller-sized LMs capable of in-context learning are created using methods like pattern-exploiting training (PET, Schick and Schütze, 2021a,b) and instruction-finetuning (T0, Sanh et al., 2022; FLAN, Chung et al., 2022; Tk-Instruct, Wang et al., 2022). OpenAI's text-davinci-003[2] has been trained using RLHF (reinforcement learning using human feedback). In-context learning-based dialog systems (Madotto et al., 2021) using LLMs like GPT-J (Wang and Komatsuzaki, 2021) or GPT-3 (Brown et al., 2020) have also been investigated, but it is crucial to select the right prompts and context to achieve the best results.

---

[2]https://platform.openai.com/docs/models

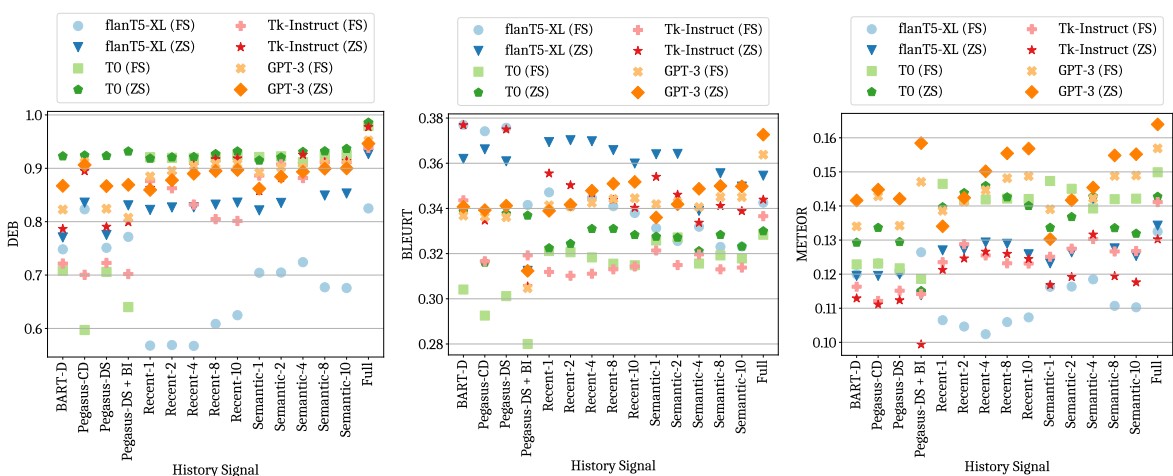

Figure 5: Performance results for Topical-Chat Dataset, Manually designed prompts. *DH = Dialog History*, *BI = Background Information*.

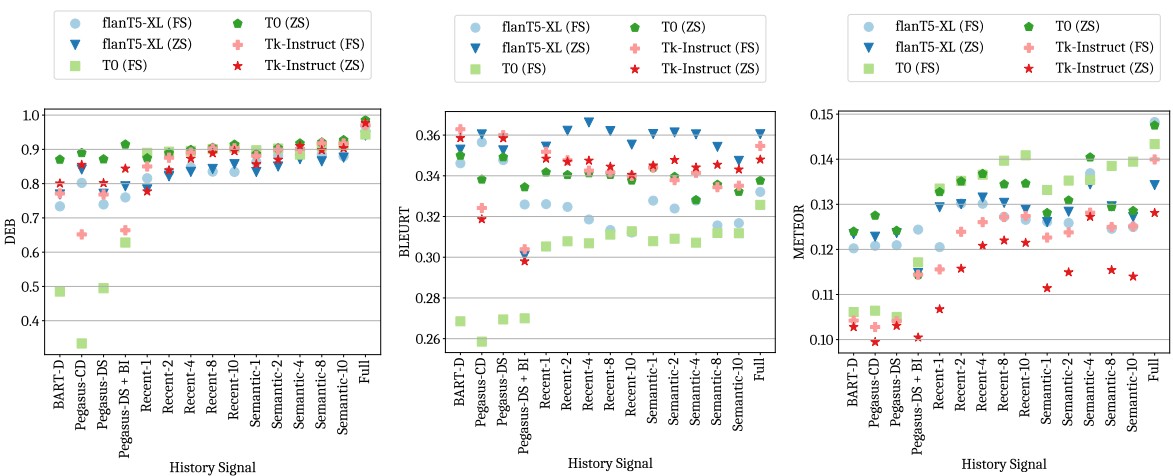

Figure 6: Performance results for Topical-Chat Dataset, Perplexity-optimized Prompts. *DH = Dialog History*, *BI = Background Information*.

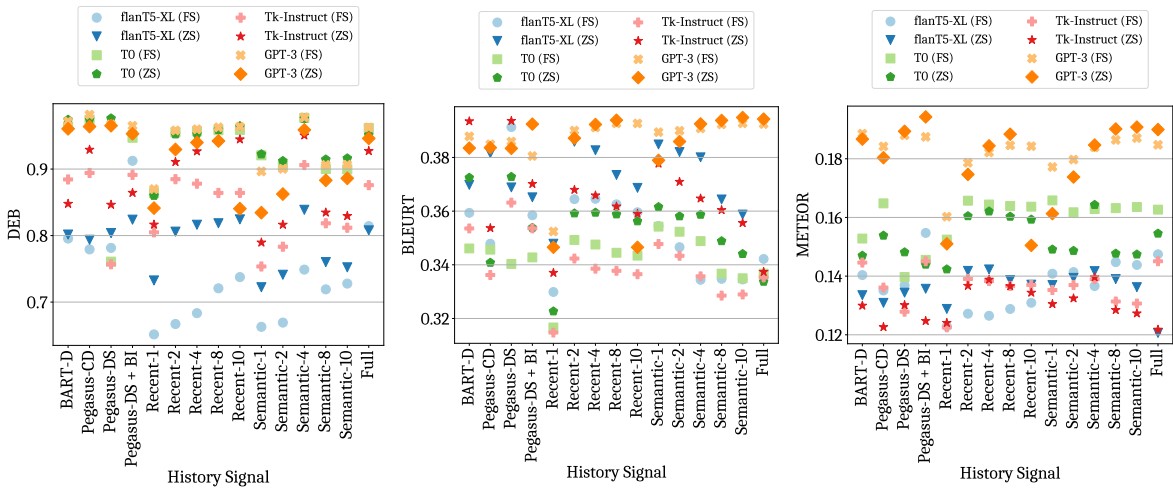

Figure 7: Performance results for Multi-Session-Chat Dataset, Manually designed prompts. *DH = Dialog History*, *BI = Background Information*.

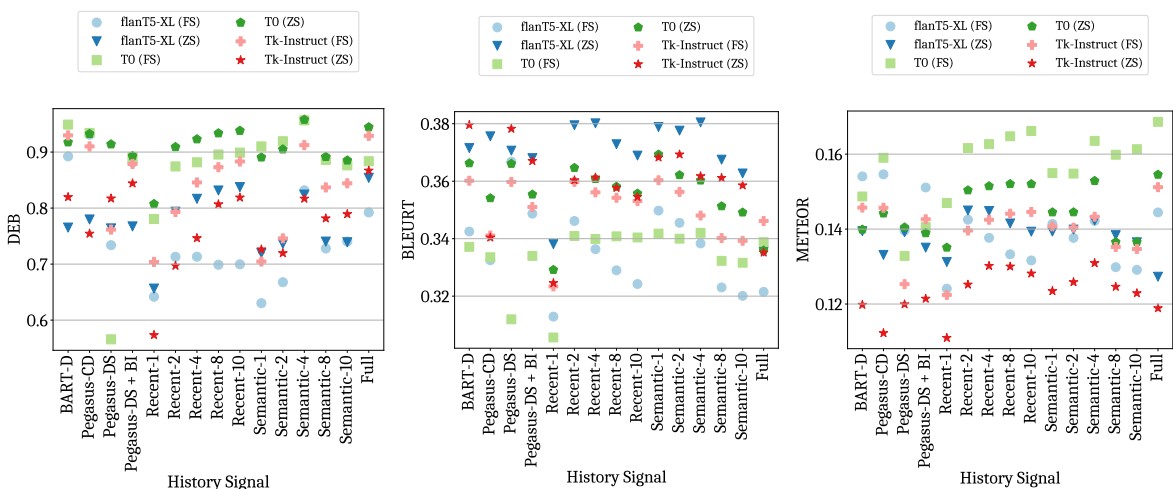

Figure 8: Performance results for Multi-Session-Chat Dataset, Perplexity-optimized Prompts. *DH = Dialog History, BI = Background Information.*

| | History | Manual Prompt | | | | | | Perplexity Prompt | | | | | |
|---|---|---|---|---|---|---|---|---|---|---|---|---|---|
| | | BLEURT | | DEB | | METEOR | | BLEURT | | DEB | | METEOR | |
| | | ZS | FS | ZS | FS | ZS | FS | ZS | FS | ZS | FS | ZS | FS |
| MSC | BART-D | 18.89 | 9.04 | 40.95 | 20.01 | 6.82 | 3.53 | 19.12 | 9.00 | 39.38 | 23.44 | 7.18 | 4.05 |
| | Pegasus-CD | 18.24 | 8.15 | 37.94 | 18.26 | 6.26 | 3.17 | 18.02 | 8.16 | 37.41 | 22.84 | 6.39 | 3.79 |
| | Pegasus-DS | 18.75 | 9.97 | 40.85 | 19.92 | 6.83 | 3.49 | 19.05 | 9.55 | 39.26 | 19.11 | 7.15 | 3.64 |
| | Pegasus-DS + BI | 11.86 | 5.89 | 26.76 | 14.99 | 4.41 | 2.54 | 11.94 | 5.72 | 24.89 | 14.45 | 4.38 | 2.48 |
| | Recent-1 | 22.61 | 9.93 | 47.60 | 19.60 | 8.37 | 3.70 | 21.47 | 9.79 | 41.68 | 20.07 | 8.33 | 3.88 |
| | Recent-2 | 19.90 | 8.88 | 41.56 | 16.25 | 7.31 | 3.10 | 19.50 | 8.61 | 40.78 | 17.73 | 7.45 | 3.54 |
| | Recent-4 | 15.59 | 7.12 | 33.25 | 13.34 | 5.80 | 2.47 | 15.47 | 6.59 | 33.23 | 13.97 | 5.89 | 2.70 |
| | Recent-8 | 11.51 | 5.39 | 25.23 | 10.71 | 4.27 | 1.92 | 11.44 | 4.86 | 25.51 | 10.33 | 4.34 | 1.97 |
| | Recent-10 | 10.37 | 4.89 | 23.19 | 10.03 | 3.86 | 1.78 | 10.33 | 4.38 | 23.46 | 9.46 | 3.90 | 1.78 |
| | Semantic-1 | 23.83 | 10.02 | 44.73 | 18.74 | 8.48 | 3.98 | 23.36 | 10.38 | 44.38 | 18.70 | 8.59 | 4.19 |
| | Semantic-2 | 20.23 | 8.44 | 39.25 | 16.30 | 7.38 | 3.44 | 20.07 | 8.85 | 39.13 | 17.10 | 7.43 | 3.52 |
| | Semantic-4 | 15.74 | 6.64 | 34.75 | 14.86 | 5.87 | 2.71 | 15.64 | 6.90 | 33.89 | 16.96 | 5.85 | 2.90 |
| | Semantic-8 | 11.07 | 4.79 | 23.10 | 10.28 | 4.22 | 2.07 | 11.21 | 4.78 | 22.58 | 10.78 | 4.22 | 1.92 |
| | Semantic-10 | 9.63 | 4.27 | 20.20 | 9.28 | 3.66 | 1.83 | 9.79 | 4.21 | 19.95 | 9.73 | 3.68 | 1.70 |
| | Full | 6.77 | 3.33 | 16.26 | 7.92 | 2.43 | 1.44 | 6.67 | 3.16 | 16.97 | 7.78 | 2.53 | 1.42 |
| TC | BART-D | 25.65 | 12.52 | 54.60 | 24.86 | 8.47 | 3.99 | 24.43 | 11.91 | 53.19 | 25.25 | 8.54 | 4.14 |
| | Pegasus-CD | 20.18 | 10.09 | 46.06 | 22.21 | 6.59 | 3.32 | 19.80 | 9.91 | 46.24 | 22.30 | 6.75 | 3.36 |
| | Pegasus-DS | 25.44 | 12.43 | 54.65 | 24.85 | 8.46 | 3.99 | 24.30 | 11.93 | 53.12 | 25.37 | 8.52 | 4.15 |
| | Pegasus-DS + BI | 7.85 | 4.28 | 20.87 | 9.67 | 2.86 | 1.58 | 7.51 | 3.95 | 19.79 | 9.22 | 2.87 | 1.51 |
| | Recent-1 | 25.81 | 11.24 | 57.44 | 18.38 | 8.87 | 3.45 | 24.16 | 11.12 | 53.39 | 27.82 | 8.82 | 4.11 |
| | Recent-2 | 21.76 | 9.55 | 48.59 | 15.92 | 7.48 | 2.93 | 21.02 | 9.43 | 47.69 | 24.25 | 7.55 | 3.78 |
| | Recent-4 | 17.08 | 7.75 | 38.20 | 12.76 | 5.97 | 2.30 | 16.83 | 7.30 | 38.34 | 19.43 | 6.04 | 2.98 |
| | Recent-8 | 12.69 | 5.77 | 28.86 | 10.29 | 4.47 | 1.79 | 12.47 | 5.37 | 29.04 | 14.32 | 4.49 | 2.18 |
| | Recent-10 | 11.39 | 5.21 | 26.45 | 9.63 | 3.98 | 1.65 | 11.14 | 4.88 | 26.85 | 13.05 | 4.04 | 1.98 |
| | Semantic-1 | 25.44 | 10.59 | 57.39 | 22.52 | 8.61 | 3.72 | 25.17 | 11.18 | 58.22 | 29.43 | 8.80 | 4.30 |
| | Semantic-2 | 21.60 | 8.91 | 49.54 | 19.28 | 7.50 | 3.18 | 21.45 | 9.46 | 50.46 | 25.71 | 7.62 | 3.67 |
| | Semantic-4 | 15.35 | 7.35 | 40.89 | 16.03 | 6.42 | 2.62 | 16.40 | 7.50 | 39.65 | 20.53 | 6.12 | 3.13 |
| | Semantic-8 | 12.16 | 5.25 | 29.03 | 11.01 | 4.36 | 1.80 | 12.14 | 5.33 | 29.64 | 14.94 | 4.44 | 2.10 |
| | Semantic-10 | 10.82 | 4.78 | 26.39 | 10.00 | 3.88 | 1.63 | 10.73 | 4.85 | 27.07 | 13.46 | 3.93 | 1.91 |
| | Full | 8.61 | 4.03 | 22.50 | 9.71 | 3.26 | 1.56 | 8.70 | 4.01 | 22.69 | 11.70 | 3.24 | 1.79 |

Table 5: UID results for the FLAN-T5 model.

## F.3 Compute Intensive LLMs

One of the most critical drawbacks of these LLMs is the training and inferencing cost, especially for long sequences. Other than the complexity of a single forward pass, there are other costs involved in training an effective transformer LLM, e.g., amount of training data and compute needed (FLOP). Strubell et al. (2019) discusses the environmental impact that the training process of these LLMs has, in terms of total $CO_2$ emissions. Optimizing costs of LMs has mainly been explored from the perspective of increasing the efficiency of the inference step of a transformer. Model distillation-based (Hinton et al., 2015; Sanh et al., 2019; Gou et al., 2021; Gupta and Agrawal, 2022) methods train a smaller, simplified model to approximate the predictions of a larger, more complex model. Efficient transformer architectures, such as Reformer (Kitaev et al., 2020), Linformer (Wang et al., 2020), BigBird (Zaheer et al., 2020), and Longformer (Beltagy et al., 2020), aim to reduce the quadratic complexity of the standard transformer architecture by using more efficient self-attention mechanisms.

In this paper, we examine the costs associated with the use of Large Language Model (LLMs)

| | History | Manual Prompt | | | | | | Perplexity Prompt | | | | | |
|---|---|---|---|---|---|---|---|---|---|---|---|---|---|
| | | BLEURT | | DEB | | METEOR | | BLEURT | | DEB | | METEOR | |
| | | ZS | FS | ZS | FS | ZS | FS | ZS | FS | ZS | FS | ZS | FS |
| MSC | BART-D | 18.66 | 8.63 | 48.81 | 24.03 | 7.36 | 3.81 | 21.18 | 8.48 | 53.06 | 23.88 | 8.09 | 3.74 |
| | Pegasus-CD | 16.26 | 7.96 | 46.51 | 22.45 | 7.34 | 3.80 | 18.93 | 7.85 | 49.86 | 21.97 | 7.71 | 3.74 |
| | Pegasus-DS | 18.59 | 8.46 | 48.67 | 18.91 | 7.39 | 3.47 | 20.98 | 7.84 | 52.38 | 14.21 | 8.05 | 3.34 |
| | Pegasus-DS + BI | 11.36 | 5.55 | 30.66 | 15.35 | 4.62 | 2.36 | 11.73 | 5.33 | 29.48 | 14.21 | 4.59 | 2.24 |
| | Recent-1 | 20.94 | 9.34 | 55.82 | 25.49 | 9.24 | 4.50 | 23.80 | 8.90 | 58.37 | 22.72 | 9.77 | 4.28 |
| | Recent-2 | 18.19 | 8.34 | 48.24 | 22.80 | 8.13 | 3.96 | 20.49 | 8.01 | 51.08 | 20.54 | 8.45 | 3.80 |
| | Recent-4 | 14.13 | 6.65 | 37.46 | 18.30 | 6.37 | 3.15 | 15.53 | 6.41 | 39.72 | 16.64 | 6.52 | 3.07 |
| | Recent-8 | 10.60 | 5.04 | 28.32 | 14.03 | 4.73 | 2.40 | 11.30 | 4.93 | 29.45 | 12.95 | 4.80 | 2.38 |
| | Recent-10 | 9.63 | 4.61 | 26.07 | 12.88 | 4.31 | 2.20 | 10.15 | 4.52 | 26.76 | 11.93 | 4.34 | 2.21 |
| | Semantic-1 | 22.40 | 9.81 | 57.14 | 25.49 | 9.24 | 4.59 | 25.37 | 9.38 | 61.17 | 24.97 | 9.93 | 4.25 |
| | Semantic-2 | 18.85 | 8.49 | 48.01 | 21.82 | 7.82 | 3.90 | 20.93 | 8.12 | 52.31 | 21.94 | 8.35 | 3.70 |
| | Semantic-4 | 14.40 | 6.88 | 39.18 | 19.26 | 6.60 | 3.21 | 15.92 | 6.62 | 42.33 | 18.51 | 6.75 | 3.17 |
| | Semantic-8 | 10.43 | 4.78 | 27.34 | 12.78 | 4.42 | 2.32 | 11.16 | 4.70 | 28.32 | 12.52 | 4.34 | 2.26 |
| | Semantic-10 | 9.10 | 4.23 | 24.23 | 11.37 | 3.90 | 2.07 | 9.76 | 4.17 | 24.74 | 11.03 | 3.82 | 2.03 |
| | Full | 6.32 | 3.27 | 18.01 | 9.34 | 2.93 | 1.58 | 6.54 | 3.33 | 18.39 | 8.67 | 3.01 | 1.65 |
| TC | BART-D | 23.04 | 9.68 | 62.76 | 22.55 | 8.79 | 3.91 | 26.50 | 8.79 | 65.90 | 15.87 | 9.39 | 3.47 |
| | Pegasus-CD | 17.03 | 7.63 | 49.80 | 15.58 | 7.20 | 3.21 | 20.07 | 6.91 | 52.80 | 8.91 | 7.57 | 2.84 |
| | Pegasus-DS | 22.81 | 9.56 | 62.37 | 22.40 | 8.75 | 3.86 | 26.23 | 8.78 | 65.47 | 16.13 | 9.33 | 3.42 |
| | Pegasus-DS + BI | 7.74 | 3.44 | 21.39 | 7.87 | 2.64 | 1.46 | 7.78 | 3.23 | 21.27 | 7.51 | 2.66 | 1.40 |
| | Recent-1 | 22.30 | 10.17 | 63.52 | 29.15 | 9.66 | 4.64 | 25.65 | 9.55 | 65.64 | 27.83 | 9.96 | 4.18 |
| | Recent-2 | 18.68 | 8.86 | 53.01 | 25.42 | 8.28 | 3.92 | 21.37 | 8.34 | 55.82 | 24.20 | 8.48 | 3.66 |
| | Recent-4 | 14.64 | 7.05 | 40.75 | 20.37 | 6.45 | 3.14 | 16.39 | 6.69 | 43.15 | 19.53 | 6.57 | 2.98 |
| | Recent-8 | 10.95 | 5.27 | 30.66 | 15.36 | 4.72 | 2.37 | 11.95 | 5.14 | 31.75 | 14.87 | 4.72 | 2.31 |
| | Recent-10 | 9.90 | 4.80 | 28.09 | 14.09 | 4.22 | 2.16 | 10.69 | 4.71 | 28.92 | 13.65 | 4.26 | 2.12 |
| | Semantic-1 | 22.63 | 10.17 | 63.22 | 28.76 | 9.23 | 4.60 | 25.98 | 9.53 | 66.90 | 27.78 | 9.67 | 4.12 |
| | Semantic-2 | 18.83 | 8.83 | 52.98 | 24.89 | 7.87 | 3.91 | 21.39 | 8.23 | 56.92 | 24.03 | 8.25 | 3.60 |
| | Semantic-4 | 14.62 | 6.96 | 42.36 | 20.10 | 6.50 | 3.07 | 15.98 | 6.65 | 44.69 | 19.15 | 6.84 | 2.93 |
| | Semantic-8 | 10.91 | 5.15 | 30.98 | 14.91 | 4.44 | 2.29 | 11.80 | 4.98 | 32.35 | 14.50 | 4.55 | 2.21 |
| | Semantic-10 | 9.70 | 4.67 | 28.12 | 13.59 | 3.96 | 2.09 | 10.48 | 4.53 | 29.25 | 13.27 | 4.05 | 2.03 |
| | Full | 7.76 | 3.86 | 23.18 | 11.52 | 3.36 | 1.76 | 8.08 | 3.89 | 23.56 | 11.28 | 3.53 | 1.71 |

Table 6: UID results for the T0 model.

| | History | Manual Prompt | | | | | | Perplexity Prompt | | | | | |
|---|---|---|---|---|---|---|---|---|---|---|---|---|---|
| | | BLEURT | | DEB | | METEOR | | BLEURT | | DEB | | METEOR | |
| | | ZS | FS | ZS | FS | ZS | FS | ZS | FS | ZS | FS | ZS | FS |
| MSC | BART-D | 21.99 | 8.89 | 47.38 | 22.25 | 7.26 | 3.64 | 20.51 | 9.35 | 44.30 | 24.13 | 6.48 | 3.78 |
| | Pegasus-CD | 17.95 | 7.89 | 47.15 | 20.99 | 6.22 | 3.19 | 16.30 | 8.19 | 36.13 | 21.84 | 5.38 | 3.50 |
| | Pegasus-DS | 22.13 | 9.27 | 47.58 | 19.32 | 7.32 | 3.26 | 20.62 | 9.32 | 44.55 | 19.73 | 6.54 | 3.25 |
| | Pegasus-DS + BI | 12.82 | 5.83 | 29.94 | 14.71 | 4.32 | 2.39 | 12.23 | 5.66 | 28.13 | 14.17 | 4.05 | 2.30 |
| | Recent-1 | 22.64 | 9.52 | 54.86 | 24.34 | 8.34 | 3.71 | 21.61 | 10.09 | 38.17 | 21.97 | 7.39 | 3.82 |
| | Recent-2 | 19.48 | 8.30 | 48.21 | 21.46 | 7.24 | 3.37 | 18.95 | 8.95 | 36.66 | 19.73 | 6.58 | 3.47 |
| | Recent-4 | 15.24 | 6.57 | 38.59 | 17.05 | 5.78 | 2.68 | 14.85 | 7.05 | 30.68 | 16.74 | 5.35 | 2.82 |
| | Recent-8 | 11.32 | 5.01 | 29.50 | 12.82 | 4.28 | 2.03 | 11.02 | 5.33 | 24.85 | 13.13 | 4.00 | 2.17 |
| | Recent-10 | 10.26 | 4.58 | 27.01 | 11.75 | 3.84 | 1.86 | 9.95 | 4.87 | 22.98 | 12.17 | 3.60 | 1.99 |
| | Semantic-1 | 24.03 | 9.92 | 50.24 | 21.50 | 8.30 | 3.86 | 23.96 | 10.62 | 47.26 | 20.77 | 8.03 | 4.15 |
| | Semantic-2 | 20.14 | 8.47 | 44.33 | 19.32 | 7.19 | 3.38 | 20.25 | 9.07 | 39.46 | 18.99 | 6.90 | 3.57 |
| | Semantic-4 | 15.63 | 6.70 | 40.74 | 18.08 | 5.98 | 2.77 | 15.31 | 7.09 | 34.57 | 18.58 | 5.54 | 2.92 |
| | Semantic-8 | 11.11 | 4.79 | 25.74 | 11.93 | 3.96 | 1.91 | 11.21 | 5.06 | 24.27 | 12.45 | 3.87 | 2.01 |
| | Semantic-10 | 9.71 | 4.27 | 22.65 | 10.54 | 3.48 | 1.70 | 9.85 | 4.49 | 21.68 | 11.16 | 3.38 | 1.78 |
| | Full | 6.86 | 3.29 | 18.84 | 8.60 | 2.47 | 1.42 | 6.78 | 3.44 | 17.54 | 9.23 | 2.41 | 1.50 |
| TC | BART-D | 27.22 | 11.31 | 56.80 | 23.75 | 8.15 | 3.83 | 26.00 | 12.30 | 58.05 | 26.14 | 7.45 | 3.53 |
| | Pegasus-CD | 19.09 | 8.52 | 51.04 | 18.85 | 6.34 | 3.02 | 17.83 | 8.81 | 47.84 | 17.71 | 5.57 | 2.79 |
| | Pegasus-DS | 26.97 | 11.18 | 56.85 | 23.70 | 8.08 | 3.78 | 25.88 | 12.17 | 57.89 | 25.97 | 7.44 | 3.52 |
| | Pegasus-DS + BI | 7.66 | 4.01 | 20.04 | 8.81 | 2.49 | 1.43 | 7.49 | 3.70 | 21.22 | 8.09 | 2.53 | 1.39 |
| | Recent-1 | 25.13 | 10.07 | 61.05 | 28.29 | 8.57 | 3.99 | 25.13 | 11.81 | 56.07 | 28.54 | 7.70 | 3.88 |
| | Recent-2 | 21.15 | 8.66 | 53.67 | 24.08 | 7.52 | 3.60 | 21.35 | 10.04 | 51.63 | 25.29 | 7.12 | 3.58 |
| | Recent-4 | 16.50 | 6.98 | 43.09 | 18.67 | 6.03 | 2.82 | 16.78 | 7.88 | 42.15 | 20.48 | 5.83 | 2.90 |
| | Recent-8 | 12.19 | 5.32 | 32.51 | 13.67 | 4.47 | 2.09 | 12.31 | 5.90 | 31.76 | 15.59 | 4.36 | 2.20 |
| | Recent-10 | 10.96 | 4.87 | 29.58 | 12.42 | 4.01 | 1.91 | 11.03 | 5.35 | 29.00 | 14.22 | 3.94 | 2.01 |
| | Semantic-1 | 24.85 | 10.30 | 60.19 | 28.42 | 8.20 | 4.01 | 25.29 | 11.48 | 62.78 | 29.42 | 8.16 | 4.09 |
| | Semantic-2 | 20.68 | 8.66 | 52.68 | 24.96 | 7.12 | 3.51 | 21.46 | 9.67 | 53.67 | 25.73 | 7.09 | 3.54 |
| | Semantic-4 | 15.54 | 7.10 | 43.10 | 19.57 | 6.13 | 2.89 | 16.54 | 7.78 | 43.75 | 20.52 | 6.12 | 2.92 |
| | Semantic-8 | 11.77 | 5.13 | 31.43 | 14.93 | 4.12 | 2.08 | 12.10 | 5.61 | 31.43 | 15.38 | 4.04 | 2.10 |
| | Semantic-10 | 10.58 | 4.68 | 28.52 | 13.60 | 3.67 | 1.89 | 10.84 | 5.10 | 28.55 | 13.97 | 3.60 | 1.90 |
| | Full | 8.45 | 4.01 | 24.00 | 11.17 | 3.20 | 1.68 | 8.64 | 4.29 | 24.26 | 11.73 | 3.18 | 1.69 |

Table 7: UID results for the Tk-Instruct model.

and suggest new metrics for assessing the cost-performance trade-offs involved, as well as strategies for optimizing the inference cost with regard to the inputs.

# G Full list of prompts

## G.1 Manually engineering prompts

In this section, we provide a full list of manually engineering prompts. Tables 9 to 14 show prompt instances for six different settings: zero-shot versus few-shot, and passing persona versus dialog history summary versus both as context. The generations

| | History | Manual Prompt | | | | | |
|---|---|---|---|---|---|---|---|
| | | BLEURT | | DEB | | METEOR | |
| | | ZS | FS | ZS | FS | ZS | FS |
| MSC | BART-D | 19.60 | 9.57 | 49.10 | 23.99 | 9.54 | 4.65 |
| | Pegasus-CD | 18.59 | 8.86 | 46.71 | 22.60 | 8.74 | 4.24 |
| | Pegasus-DS | 19.40 | 9.54 | 48.84 | 23.90 | 9.58 | 4.65 |
| | Pegasus-DS + BI | 12.86 | 6.22 | 31.24 | 15.78 | 6.37 | 3.07 |
| | Recent-1 | 22.70 | 10.51 | 55.12 | 25.94 | 9.89 | 4.78 |
| | Recent-2 | 19.97 | 9.36 | 47.94 | 22.99 | 9.01 | 4.41 |
| | Recent-4 | 15.77 | 7.52 | 37.78 | 18.45 | 7.41 | 3.50 |
| | Recent-8 | 11.95 | 5.79 | 28.60 | 14.19 | 5.72 | 2.72 |
| | Recent-10 | 9.91 | 5.31 | 24.05 | 13.03 | 4.31 | 2.49 |
| | Semantic-1 | 23.76 | 10.98 | 52.33 | 25.27 | 10.12 | 5.00 |
| | Semantic-2 | 20.22 | 9.51 | 45.19 | 21.97 | 9.10 | 4.39 |
| | Semantic-4 | 16.21 | 7.71 | 39.61 | 19.29 | 7.63 | 3.63 |
| | Semantic-8 | 11.67 | 5.66 | 26.16 | 13.08 | 5.63 | 2.69 |
| | Semantic-10 | 10.42 | 5.06 | 23.38 | 11.69 | 5.03 | 2.41 |
| | Full | 7.92 | 3.87 | 19.01 | 9.48 | 3.82 | 1.82 |
| TC | BART-D | 22.34 | 10.92 | 56.84 | 26.67 | 9.28 | 4.34 |
| | Pegasus-CD | 18.07 | 8.83 | 48.28 | 23.92 | 7.71 | 3.75 |
| | Pegasus-DS | 22.27 | 10.85 | 56.57 | 26.60 | 9.28 | 4.33 |
| | Pegasus-DS + BI | 7.45 | 3.77 | 20.72 | 10.00 | 3.78 | 1.82 |
| | Recent-1 | 23.43 | 11.00 | 59.42 | 28.51 | 9.27 | 4.47 |
| | Recent-2 | 19.81 | 9.48 | 50.90 | 24.90 | 8.26 | 3.93 |
| | Recent-4 | 15.73 | 7.63 | 40.23 | 20.14 | 6.79 | 3.22 |
| | Recent-8 | 11.85 | 5.79 | 30.22 | 15.29 | 5.25 | 2.49 |
| | Recent-10 | 10.83 | 5.30 | 27.63 | 13.97 | 4.83 | 2.29 |
| | Semantic-1 | 23.17 | 10.93 | 59.43 | 28.51 | 8.98 | 4.44 |
| | Semantic-2 | 19.48 | 9.41 | 50.39 | 24.74 | 8.07 | 3.89 |
| | Semantic-4 | 15.84 | 7.52 | 40.58 | 19.88 | 6.61 | 3.14 |
| | Semantic-8 | 11.43 | 5.63 | 29.37 | 14.83 | 5.06 | 2.43 |
| | Semantic-10 | 10.40 | 5.13 | 26.76 | 13.50 | 4.61 | 2.21 |
| | Full | 8.99 | 4.34 | 22.82 | 11.36 | 3.95 | 1.87 |

Table 8: UID results for the GPT-3 model. Manual prompt only, since prompts for GPT-3 cannot be perplexity optimized.

are from the GPT3 model. Rather than the persona, when we use knowledge facts, the prompt templates remain the same. When the dialog context consists of components other than summary, e.g., full history or recent-$k$ utterances or semantic-$k$ utterances, "summary" in prompt templates is replaced with "full history" or "list of recent-$k$ utterances" or "list of semantic-$k$ utterances" respectively.

### G.2  Perplexity optimized prompts

Since we have only API access to the GPT3 model, we could perform perplexity optimization for only Flan T5 XL, T0 and Tk-Instruct models. Tables 15 to 19 show perplexity optimized prompts (templates as well as instance) for FlanT5XL, T0 and Tk-Instruct models under various settings like (a) zero shot versus few shot, and (b) persona, summary, knowledge section or combinations as dialog context.

## H  Impact of varying metric-importance index ($a$)

We vary $a$ as [0.5, 1, 2, 5, 10]. This updated formulation of the UID metric, with $M_H$ raised to an exponent "$a$", can be used to capture the importance assigned by the user on the model performance $M_H$, e.g., when inference cost is less of a

bottleneck. We analyzed the accuracy-length trade-off using different values for the parameter "$a$" to capture various types of user requirements in terms of the allowed expenses towards the inference process. The average UID values (for zero-shot manual prompts) across all the models are shown in Tables 20 and 21. Based on these experiments, we found the following insightful observations. These tables show that for both MSC and TC, for DEB and METEOR, as the value of "$a$" is increased, summary-based dialog history variants tend to become better in terms of UID while Recent-$k$ and Semantic-$k$ variants tend to become less impressive. Although, in terms of BLEURT (UID), the ranking is in favour of Semantic-1 or 2 and Recent-1 or 2 throughout the complete range of "$a$" that we have explored. This might be because BLEURT measures normal sentence semantic similarity but not context-response relevance as measured by DEB.

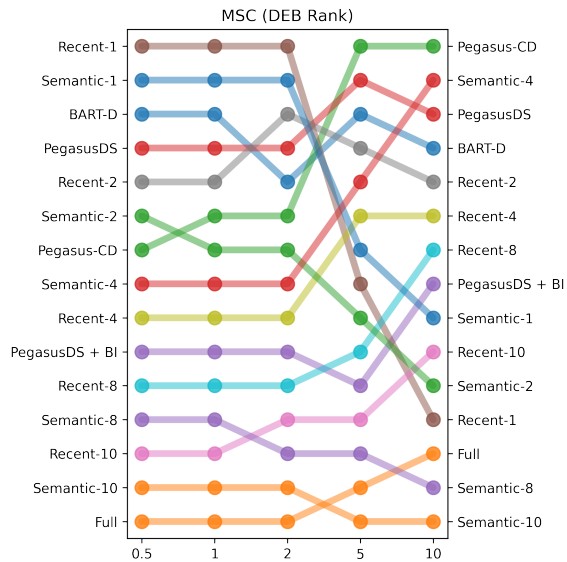

Figure 9: Trend in Ranks of History Signal Types for Different Values of the Metric-Importance Index $a$ (for MSC dataset, DEB metric)

To fully understand how the rank of various history signals vary over the value of the metric-importance "$a$", we plot the rank-order of all history signal types vs. the value of "$a$" (increased from 0.5 to 10) as show in Fig. 9. This rank order dynamics helps us clearly understand, as we give more and more importance to the model performance and ignore the cost of inference, how the choices over the history signal change. For example, in terms of the UID (DEB) metric on the MSC dataset, the average trend across models is that Recent-1 and Semantic-1 are the recom-

| Automated Chat System: |
|---|
| **This is a summary of a dialog exchange between Person1 and Person2:** Person1 and Person2 talk about the left handed QB in the NFL and the origins of the huddle. They also talk about Rush and the law that makes people listen to 40% Canadian music. |
| **Given the summary of the dialog exchange between Person1 and Person2, give a consistent and diverse response to the following dialog by Person1.** 
 **Person1**: He was a terrible hockey player, so he took up music. Would you listen to a news broadcast in Latin? 
 **Person2**: |
| Generation: Sure, I'd be happy to listen to a news broadcast in Latin. Latin is a great language with a rich history and culture, so it would be very interesting to hear a news broadcast in that language. Additionally, I'm always open to exploring new cultures and languages, so this would be a great opportunity to learn more. |

Table 9: Manually engineered prompt with summary of dialog history as dialog context and without exemplars or background information.

| Automated Chat System: |
|---|
| **Learn from the below example on how to generate consistent and diverse responses between Person1 and Person2 given summary. Example:** |
| **This is a summary of a dialog exchange between Person1 and Person2:** Person1 and Person2 are talking about Nintendo. They didn't know Nintendo was over 100 years old and that they started out as a playing card company. Person2 also tells Person1 the Army uses Xbox 360 controllers and Person1 says humans play about 3 billion hours of video games a week. |
| **Given the summary of the dialog exchange between Person1 and Person2, give a consistent and diverse response to the following dialog by Person1.** 
 **Person1:** Yeah that would have been a problem for me as a kid haha. Do you know who Stephen Fry is? He did the narration for a lot of movies and video games like the Harry Potter video games. 
 **Person2:** Yeah. He also did the narration for Little Big Planet. |
| **Now try it yourself:** |
| **This is a summary of a dialog exchange between Person1 and Person2:** Person1 tells Person2 about a Notre Dame quarterback named Ian Book. |
| **Given the summary of the dialog exchange between Person1 and Person2, give a consistent and diverse response to the following dialog by Person1.** 
 **Person1:** the chance to play for notre dame must be great. they love football there. 
 **Person2:** |
| Generation: Absolutely! Notre Dame has a rich football history and a huge fan base. It's a huge honour for any player to have the chance to play for them. |

Table 10: Manually engineered prompt with summary of dialog history but no background information as dialog context and with one exemplar.

| Automated Chat System: |
|---|
| **Here are some background details about Person1:** Person1 likes the Lord of the Rings movies and riding a bike, while Person1 is a travel agent, listening to music to relax, and Person1 could not be a vegetarian. |
| **Here are some background details about Person2:** Person2 likes to talk about Person2's favorite books, characters, and places. Person2 runs a charity and Person2's parents are doctors. |
| **Given the background details of Person1 and Person2 , give a consistent and diverse response to the following dialog spoken by Person1.** 
 **Person1**: WHAT kind of music do you listen to when you are relaxing? 
 **Person2:** |
| Generation: I usually listen to a variety of music when I'm relaxing, ranging from classical to pop, jazz, and even some hip hop. I'm always looking to discover something new and interesting. |

Table 11: Manually engineered prompt with persona as dialog context but no summary of dialog history and with no exemplars.

| Automated Chat System: |
|---|
| **Learn from the below example on how to use background details to generate a consistent and diverse response by Person2 on what Person1 says. Example:** |
| **Here are some background details about Person1**: Person1 loves ice cream and works at a fast food place. Person1 is in high school and plans to become an eagle scout. Person1 would love to volunteer at the church. |
| **Here are some background details about Person2:** Person2 recently gave birth to twins and wants many more children. Person2 is a stay-at-home mom who plays tennis and goes to the church. |
| **Given the background details of Person1 and Person2 , give a consistent and diverse response to the following dialog spoken by Person1.** 
 **Person1**: THAT's a lot of kids! BUT since you love the church so much having enough for a church choir sounds like a good idea! HAVE you had any luck finding volunteers? WHEN is your next retreat? 
 **Person2**: WE've gotten some volunteers, but always need more! THANK you for getting your scout troop involved. WE are having another retreat next month. IT'll be a youth retreat if you and your friends would be interested. |
| **Now try it yourself:** |
| **Here are some background details about Person1:** Person1 recently gave birth to twins and wants many more children. Person1 is a stay-at-home mom who plays tennis and goes to the church. |
| **Here are some background details about Person2:** Person2 loves ice cream and works at a fast food place. Person2 is in high school and plans to become an eagle scout. Person2 would love to volunteer at the church. |
| **Given the background details of Person1 and Person2 , give a consistent and diverse response to the following dialog spoken by Person1.** 
 **Person1**: WE've gotten some volunteers, but always need more! THANK you for getting your scout troop involved. WE are having another retreat next month. IT'll be a youth retreat if you and your friends would be interested. 
 **Person2:** |
| Generation: That's great! I'm sure your scout troop will be excited to help out at the retreat. I'd love to see if I can get some of my friends involved too. I'm sure they'd be interested in attending the retreat as well. |

Table 12: Manually engineered prompt with persona as dialog context but no summary of dialog history and with one exemplar.

mended ways to summarize the context information if cost is an important factor to the user. Whereas, if cost is of less importance, then longer dialog summaries such as Pegasus-CD and Semantic-4 are recommended approaches for MSC. Additionally, we observe that some models like Recent-10, Semantic-8, Semantic-10 and Full are always bad choices, while some (BART-D, Pegasus-DS) are

| Automated Chat System: |
| --- |
| **Here are some background details about Person1:** Person1 likes flying kites and works as a zoo keeper. Person1 has a bearded dragon and a pet raccoon. Person1 recommends that people adopt raccoons. |
| **Here are some background details about Person2:** Person2's a self-employed artist. Person2's looking for animals to adopt. They like the idea of getting a pet raccoon, Person2 has a dog. |
| **This is a summary of a dialog exchange between Person1 and Person2:** Person2 likes taking his dog for a work sometimes. Person1 has a bearded dragon and a racoon as zoo keepers. Person1 recommends a pet racoon to Person2. |
| **Given the background details and the summary of the dialog exchange between Person1 and Person2, give a consistent and diverse response to the following dialog by Person1.** 
 **Person1**: MY pet raccoon just learned a new trick! 
 **Person2**: |
| **Generation:** Wow! What kind of trick did your pet raccoon learn? I'm always interested in learning new things that my pet can do! |

Table 13: Manually engineered prompt with persona and summary of dialog history as dialog context and with no exemplars.

| Automated Chat System: |
| --- |
| **Learn from the below example on how to generate consistent and diverse responses between Person1 and Person2 given background details along with summary. Example:** |
| **Here are some background details about Person1:** Person1 loves ice cream and works at a fast food place. Person1 is in high school and plans to become an eagle scout. Person1 would love to volunteer at the church. |
| **Here are some background details about Person2:** Person2 recently gave birth to twins and wants many more children. Person2 is a stay-at-home mom who plays tennis and goes to the church. |
| **This is a summary of a dialog exchange between Person1 and Person2:** Person2 plays tennis every week and goes to church. Person1 works at a fast food place and loves ice cream. Person2 has twins and wants to start a choir. Person1 hopes to become an eagle scout. |
| **Given the background details and the summary of the dialog exchange between Person1 and Person2, give a consistent and diverse response to the following dialog by Person1.** 
 **Person1**: THAT's a lot of kids! BUT since you love the church so much having enough for a church choir sounds like a good idea! HAVE you had any luck finding volunteers? WHEN is your next retreat? 
 **Person2**: WE've gotten some volunteers, but always need more! THANK you for getting your scout troop involved. WE are having another retreat next month. IT'll be a youth retreat if you and your friends would be interested. |
| **Now try it yourself:** |
| **Here are some background details about Person1:** Person1 recently gave birth to twins and wants many more children. Person1 is a stay-at-home mom who plays tennis and goes to the church. |
| **Here are some background details about Person2:** Person2 loves ice cream and works at a fast food place. Person2 is in high school and plans to become an eagle scout. Person2 would love to volunteer at the church. |
| **This is a summary of a dialog exchange between Person1 and Person2:** Person2 works at a fast food place and loves ice cream. Person1 is a stay-at-home mom and wants to start her own choir. Person2 is an eagle scout and manages school and work. Person1 is starting to get back to tennis after having twins. |
| **Given the summary of the dialog exchange between Person1 and Person2 and their background details, give a consistent and diverse response to the following dialog spoken by Person1.** 
 **Person1**: WE've gotten some volunteers, but always need more! THANK you for getting your scout troop involved. WE are having another retreat next month. IT'll be a youth retreat if you and your friends would be interested. 
 **Person2**: |
| **Generation:** That sounds great! I would love to volunteer for the retreat and get my friends involved as well. How can I help? What kind of activities do you have planned? |

Table 14: Manually engineered prompt with persona and summary of dialog history as dialog context and with one exemplar.

quite robust across the whole range of values of
"$a$".

| Setting | Prompt |
|---|---|
| Only Summary Few Shot | **Take a look at the following example for guidance. Example: Here is a conversation summary between Person1 and Person2:** Person2 and Person1 talk about their hobbies. Person2 has a big family and Person2 likes to hang out with Person2's mom on her days off from fedex. Person1 has twins and enjoys playing tennis. They also talk about other hobbies they might want to take up for their kids. **Based on the summary of conversation between Person1 and Person2, what do you think Person2 will say next? Person1:** WELL I guess I can try swimming. WHAT are your tips when you want to try something but are afraid of it? **Person2:** I think the most important thing to remember is that it's okay if you fail. YOU can always get back up and try again. I do that when I play tennis but I do it with everything else I do, too. I like cooking. I like basketball. I even like playing video games. I always just try to have fun and if I fail, I try again. I know you like tennis but what else do you like to do? **Now try it yourself. Here is a conversation summary between Person1 and Person2:** Person2 and Person1 talk about their hobbies. Person1 has 2 brothers and 2 sisters. Person2 has a big family and Person2 enjoys shopping, going to the movies, and playing tennis. Then they talk about other hobbies they might want to take up for their kids. **Based on the summary of conversation between Person1 and Person2, what do you think Person2 will say next? Person1:** I think the most important thing to remember is that it's okay if you fail. YOU can always get back up and try again. I do that when I play tennis but I do it with everything else I do, too. I like cooking. I like basketball. I even like playing video games. I always just try to have fun and if I fail, I try again. I know you like tennis but what else do you like to do? **Person2:** I like cooking. I like basketball. I even like playing video games. I always just try to have fun and if I fail, I try again. I know you like tennis but what else do you like to do? |
| Only Summary Zero Shot | **Here is a summary of the conversation between Person1 and Person2:** Person2 and Person1 talk about their jobs in the food industry. Person1 works at a car dealership in sales and likes music. Person2 likes 21 pilots and Person2 got the new 2021 subaru as Person2's dream car. **Based on the dialogue between the Person1 and the Person2 so far, try to anticipate what the Person2's response might be to the Person1's next statement. Person1:** YEAH they are really nice cars. HOW much did you pay for it after? **Person2:** I hope to over the summer. I just got out of school. |
| Only Persona Few Shot | **Learn from the example first. Example: Here is some information about the Person1 and Person2:** Person2 has been to Boston but grew up in San Francisco. Person2 works a lot and fishs in his spare time. Person2 is married with children. Person1 is an English teacher in Boston. Person1 likes drawing and Fish 'n Chips. Person1 like Batman and Fish 'n Chips. **Based on the information provided about the Person1 and the Person2, predict what might have been said following this dialogue by the Person1. Person1:** I went fishing yesterday. I pulled enough in to have dinner. **Person2:** WOW!! THAT's great! WHO cooked? **Now try it yourself. Here is some information about the Person1 and Person2:** Person1 and Person1 are loners. Person1 likes grilling and hiking along the ocean, but Person1 is scared of swimming in the water, while Person1 likes swimming in the water. Person2 tells Person2 about Person2's personality. Person2's nervous about hearing back on Wednesday. **Based on the information provided about the Person1 and the Person2, predict what might have been said following this dialogue by the Person1. Person1:** CONGRATULATIONS! BE sure to show them that you should have been their first choice. SO more family time at the beach now? **Person2:** MY kids have been asking for a cat. I don't know if cats and water go along. |
| Only Persona Zero Shot | **Here is some information about the Person1 and Person2:** Person1 and Person1 talk about their hobbies. Person1 likes to work on computers, read, and ride a bike. Person1 is a vegetarian. Person2 is a car salesman and Person2 is a fan of the band 5 Finger Death Punch and the Slipknot song Duality and its music video. **Based on the information provided about the Person1 and the Person2, predict what the Person2 might have said in response to the Person1's dialogue. Person1:** THAT's a huge number, especially given the commissions you must get. WHAT kind of cut do you get per car? **Person2:** WE get 25% of the gross profit of the car. HOW is your job? ARE you looking for another? |
| Persona + Summary Few Shot | **Take a look at the following example for guidance. Example: Here is some information about Person1 and Person2:** Person2 and Person2 are talking about their personal characteristics. Person2 is single and works from home doing programming and web design. Person2 doesn't like birds and wants to watch North by Northwest. Person1 is single and Person1 likes conspiracy theories. Person1's favorite Hitchcock movies are Vertigo, Rear Window, and North by Northwest. Person1 doesn't like birds. **Here is a conversation summary between Person1 and Person2:** Person2 is single and doesn't want to get married. Person2 prefers to read outdoors alone. Person2 is an introvert and worries about running into wild animals. Person2 just bought north by northwest. **In the light of the conversation summary and the information provided about the Person1 and the Person2, predict what might have been said following this dialogue by the Person1. Person1:** LET me know if its a good one. I watched city of lies last night. IT was pretty good. I want to see wanderer next. **Person2:** I think most hitchcock's hold up but man he was kind of a jerk. **Now try it yourself. Here is some information about the Person1 and Person2:** Person1 and Person1 are talking about their personal characteristics. Person1 is single and works from home doing programming and web design. Person1 doesn't like birds and wants to watch North by Northwest. Person2 is single and Person2 likes conspiracy theories. Person2's favorite Hitchcock movies are Vertigo, Rear Window, and North by Northwest. Person2 doesn't like birds. **Here is a conversation summary between Person1 and Person2.** Person1 is single and doesn't want to get married. Person1 prefers to read outdoors alone. Person1 worries about running into wild animals. Person1 bought north by northwest. Person2 watched city of lies last night. **In the light of the conversation summary and the information provided about the Person1 and the Person2, predict what might have been said following this dialogue by the Person1. Person1:** I think most hitchcock's hold up but man he was kind of a jerk. **Person2:** LET me know if its a good one. I watched city of lies last night. IT was pretty good. I want to see wanderer next. |
| Persona + Summary Zero Shot | **Here are some persona details about the Person1 and Person2:** Person1 and Person1 talk about their hobbies. Person1 likes camping, leisure activities, coffee, and Corona beer. Person1 went hiking on memorial day. Person2 loves the beach. Person2 likes to cook and make coffee. Person2 lives in California and has kids. Person2 is thirty years old. **Here is a conversation summary between Person1 and Person2:** Person1's family asked her to go camping for the holiday weekend. Person2 is visiting her family over the summer. Person2 is going to Las Vegas with her mom and sisters for a weekend. **Based on the persona information of the Person1 and the Person2 and their conversation summary so far, anticipate what the Person2's response might be to the Person1's next statement. Person1:** ARE you a gambler or are you going there to people watch and go to shows? **Person2:** I'm a gambler. I'm going to the vegas strip |
| Only Knowledge Zero Shot | **Here is some data on the topics the Person1 and Person2 are discussing about:** Spider-Man first appeared in the anthology comic book Amazing Fantasy 15 (August 1962) in the Silver Age of Comic Books..In the stories, Spider-Man is the alias of Peter Parker, an orphan raised by his Aunt May and Uncle Ben in New York City after his parents Richard and Mary Parker were killed in a plane crash..His origin story has him acquiring spider-related abilities after a bite from a radioactive spider; these include clinging to surfaces, shooting spider-webs from wrist-mounted devices, Marvel Comics is the brand name and primary imprint of Marvel Worldwide Inc.. In 2009, The Walt Disney Company acquired Marvel Entertainment, Marvel Worldwide's parent company..Marvel started in 1939 as Timely Publications, and by the early 1950s, had generally become known as Atlas Comics. Comic books were first popularized in the United States and the United Kingdom during the 1930s ..The term comic book derives from American comic books once being a compilation of comic strips of a humorous tone ..The Marvel Cinematic Universe takes place in Earth-199999, a multiverse different from the original comic book universe, Earth-616. **Based on the data about the topics being discussed by the Person1 and the Person2, anticipate what the Person2 might have said next in response to the Person1's dialogue. Person1:** NAH, I usually eat a yogurt berry smoothie for breakfast. YOU? **Person2:** I usually eat a bowl of oatmeal. What do you eat for breakfast? I'm curious. |

Table 15: Perplexity Optimized Prompt Examples for FlanT5XL. Template part is shown in bold. The last output for the Person2 is the generated output using FlanT5XL. Part 1

| Setting | Prompt |
| --- | --- |
| Only Knowledge Few Shot | **Learn from the example first. Example: Here is some information on the topics being discussed by Person1 and Person2:** The name Rotten Tomatoes derives from the practice of audiences throwing rotten tomatoes when disapproving of a poor stage performance..Michael Bay's average Rotten Tomatoes film rating is 38%. no film based on a video game has achieved above 44% on rotten tomatoes..Netflix has almost 150 movies available with a 100% rating on Rotten Tomatoes. Incredibles 2 is a 2018 American computer-animated superhero film ..It is the sequel to The Incredibles (2004) and the second full-length installment of the franchise ..Craig T. Nelson, Holly Hunter, Sarah Vowell and Samuel L. Jackson reprise their roles from the first film .. newcomers to the cast include Huckleberry Milner, Bob Odenkirk, Catherine Keener and Jonathan Banks . Box office business can be measured in the terms of the number of tickets sold or the amount of money raised by ticket sales (revenue).With over $8.5 billion worldwide film earnings, Tom Hanks is the highest all-time box office star ..The Silence of the Lambs came out on Valentine's day in 1991 and had a box office of over $270 million. **Based on the data about the topics being discussed by the Person1 and the Person2, anticipate what the Person2 might have said next in response to the Person1's dialogue. Person1:** MARS is also considered a terrestrial planet and the fourth planet from the sun. MUST be hot there. **Person2:** BUT it has polar icecaps like us. **Now try it yourself. Here is some data on the topics the Person1 and Person2 are discussing about:** A planet is an astronomical body orbiting a star or stellar remnant that is massive enough to be rounded by its own gravity ..The term planet is ancient, with ties to history, astrology, science, mythology, and religion ..In 2006, the International Astronomical Union (IAU) officially adopted a resolution defining planets within the Solar System . A planet is an astronomical body orbiting a star or stellar remnant that is massive enough to be rounded by its own gravity ..The term planet is ancient, with ties to history, astrology, science, mythology, and religion ..In 2006, the International Astronomical Union (IAU) officially adopted a resolution defining planets within the Solar System . A planet is an astronomical body orbiting a star or stellar remnant that is massive enough to be rounded by its own gravity ..The term planet is ancient, with ties to history, astrology, science, mythology, and religion ..In 2006, the International Astronomical Union (IAU) officially adopted a resolution defining planets within the Solar System . **Based on the data about the topics being discussed by the Person1 and the Person2, anticipate what the Person2 might have said next in response to the Person1's dialogue. Person1:** OH man, thats sad for him. I like his movies usually. **Person2:** I don't recall too many of his movies, I've seen a couple and they are alright. I did like him in that 70s show! |
| Knowledge + Summary Few Shot | **Learn from the example first. Example: Here is some data on the topics the Person1 and Person2 are discussing about:** The Walt Disney Company was founded on October 16, 1923 by brothers Walt and Roy O. Disney as the Disney Brothers Cartoon Studio ..The company established itself as a leader in the American animation industry before diversifying into live-action film production, television, and theme parks ..It is the world's largest independent media conglomerate in terms of revenue . The film is a live-action reimagining of Disney's 1991 animated film of the same name, itself an adaptation of Jeanne-Marie Leprince de Beaumont's 18th-century fairy tale ..The film features an ensemble cast that includes Emma Watson and Dan Stevens as the eponymous characters with Luke Evans, Kevin Kline, Josh Gad, Ewan McGregor, Stanley Tucci, Audra McDonald, Gugu Mbatha-Raw, Ian McKellen, and Emma Thompson in supporting roles . Morgan Freeman won an Academy Award in 2005 for Best Supporting Actor with Million Dollar Baby ..He has received Oscar nominations for his performances in Street Smart (1987), Driving Miss Daisy (1989), The Shawshank Redemption (1994), and Invictus 2009 ..Morgan Freeman played a singing vampire obsessed with vegetables on The Electric Company in the 70's . **Here is a conversation summary between Person1 and Person2:** Person2 and Person1 are talking about the movie Deadpool 2 . They also talk about Marvel comics and the X-men. Person2 likes the X-men and Person1 likes Spider-Man. **In the light of the chat history and information about topics that the Person1 and the Person2 are chatting on, predict what might have been said following this dialogue by the Person1. Person1:** DID you know marvel successfully argues in court that mutants in x-men are not humans? **Person2:** I didnt and did winning that argument helped them in any way? **Now try it yourself. Here is some data on the topics the Person1 and Person2 are discussing about:** It is the eleventh installment in the X-Men film series, and a direct sequel to the 2016 film Deadpool ..In the film, Deadpool forms the team X-Force to protect a young mutant from the time-traveling soldier Cable ..Deadpool is the highest grossing R-rated film of all time, the highest grossing X-Men film, and the highest grossing 20th Century Fox film not directed by James Cameron or George Lucas . It is the eleventh installment in the X-Men film series, and a direct sequel to the 2016 film Deadpool ..In the film, Deadpool forms the team X-Force to protect a young mutant from the time-traveling soldier Cable ..Deadpool is the highest grossing R-rated film of all time, the highest grossing X-Men film, and the highest grossing 20th Century Fox film not directed by James Cameron or George Lucas . It is the eleventh installment in the X-Men film series, and a direct sequel to the 2016 film Deadpool ..In the film, Deadpool forms the team X-Force to protect a young mutant from the time-traveling soldier Cable ..Deadpool is the highest grossing R-rated film of all time, the highest grossing X-Men film, and the highest grossing 20th Century Fox film not directed by James Cameron or George Lucas . **Here is a conversation summary between Person1 and Person2.** Person2h Person1 and Person2 love Disney. Person1 loved the cartoon beauty and the beast, the little mermaid, lion king, some of those iconic cartoons from when Person1 was growing up. Person2 really likes little mermaid, lion king, aladdin, frozen, moana. Person1 didn't know the disney channel doesn't accept outside ads.walt disney was fired from his newspaper job for not being more creative. **In the light of the chat history and information about topics that the Person1 and the Person2 are chatting on, predict what might have been said following this dialogue by the Person1. Person1:** IT has been a good 5 for me, but I live an hour away, so its not a huge trip. **Person2:** AH cool! YOU should go more often then lol did you see the most recent beauty and the beast? |
| Knowledge + Summary Zero Shot | **Here is some information on the topics being discussed by Person1 and Person2:** Golf is a club-and-ball sport in which players use various clubs to hit balls into a series of holes on a course ..The average American golf course consumes around 312,000 gallons of water per day ..Babe Ruth was once America's most famous golfer.Samuel L. Jackson puts a golf clause in his film contracts that allows him to play golf twice a week during production . The University of Iowa's locker room for visiting football teams is completely painted pink ..The highest score ever in a football game occurred in 1916 when Georgia Tech defeated Cumberland 222-0 ..Former Partiots RB BenJarvus Green-Ellis has never fumbled the football in his NFL career . The NFL is one of the four major professional sports leagues in North America ..The NFL has no written rule against female players; women would in fact be allowed if they met the league's eligibility requirements ..New Orleans Saints cheerleaders are forbidden from eating in the same restaurant as any NFL player . **Here is a conversation summary between Person1 and Person2:** The average golf course consumes around 312,000 gallons of water per day. There is a golf course in Dubai that needs 4 million gallons of water per day. The top bowler made twice as much as the top football stars. The average engineer makes more in his lifetime that the average NFL and mlb player. The highest scoring football game ever was 222-0. **In the light of the conversation history and information about the topics being discussed by the Person1 and the Person2, predict what might have been said following this dialogue by the Person1. Person1:** THAT seems like the oddest rule. IT's been fun chatting with you! **Person2:** The average bowler makes twice as much as the top football stars. The average engineer makes more in his lifetime that the average NFL and mlb player. |

Table 16: Perplexity Optimized Prompt Examples for Flan-T5-XL. Template part is shown in bold. The last output for the Person2 is the generated output using Flan-T5-XL. Part 2

| Setting | Prompt |
|---|---|
| Summary Few Shot | **Take a look at the following example for guidance. Example: Here is a conversation summary between Person1 and Person2:** Person1 is a sales manager for a premium mattress retailer. Person2 works in the tech industry working on software solutions. They talk about their hobbies and work. **Based on the conversation between the Person1 and the Person2, predict what the Person2 might have said in response to the Person1's dialogue. Person1:** I am glad too. THE online competition was killing us. THEN my boss decided to also sell online so we been very busy since then. **Person2:** IF you can't beat them, join them. SOME people prefer the convenience of being able to stay home and buy. **Now try it yourself. Here is a conversation summary between Person1 and Person2:** Person2 is a sales manager for a premium mattress retailer. Person1 works in the tech industry working on software solutions. They talk about their hobbies, work, and their current situation. **Based on the conversation between the Person1 and the Person2, predict what the Person2 might have said in response to the Person1's dialogue. Person1:** IF you can't beat them, join them. SOME people prefer the convenience of being able to stay home and buy. **Person2:** IF you can't beat them, join them. SOME people prefer the convenience of being able to stay home and buy. |
| Summary Zero Shot | **Here is a summary of the conversation between Person1 and Person2:** Person2 and Person1 talk about the reasons why Person1 makes terrible choices. Person1 tries to love Person2 but it's hard. Person2 tells Person1 Person2's mom loves Person1 and Person2 is planning a surprise party for Person1's mom who has mental health issues. **Based on the summary of conversation between Person1 and Person2, what do you think Person2 will say next? Person1:** OH okay, so have you done anything fun lately? **Person2:** I hope to over the summer. I just got out of school. |
| Only Persona Few Shot | **Learn from the example first. Example: Here is some information about the Person1 and Person2:** Person2 is in a band and plays jazz music. Person2 is in high school and plays jazz music frequently. Person2 thinks the bass is the easiest instrument to learn. Person1 likes jazz, snowboarding, and eating steak. Person1 wants to attend a jazz concert, and Person1 wants to learn to play an instrument. **Based on the information provided about the Person1 and the Person2, predict what the Person2 might have said in response to the Person1's dialogue. Person1:** WOW that's seriously so awesome! IT's really coming together! IF you need anyone to play some music on your opening day, let me know lol. **Person2:** WOULD you be down to play a show on opening day? THAT would actually be awesome and I think would draw a huge crowd. NOTHING beats ice cream and a show! **Now try it yourself. Here is some information about the Person1 and Person2:** Person1 is in a band and plays jazz music. Person1 is in high school and plays jazz music frequently. Person1 thinks the bass is the easiest instrument to learn. Person2 likes jazz, snowboarding, and eating steak. Person2 wants to attend a jazz concert, and Person2 wants to learn to play an instrument. **Based on the information provided about the Person1 and the Person2, predict what the Person2 might have said in response to the Person1's dialogue. Person1:** WOULD you be down to play a show on opening day? THAT would actually be awesome and I think would draw a huge crowd. NOTHING beats ice cream and a show! **Person2:** YEAH for sure! I'd love to do that. AS long as I get some steak haha I love that you're selling steak out of an ice cream truck. SUCH a unique idea! |
| Only Persona Zero Shot | **Here are some persona details about the Person1 and Person2:** Person1 is a landlord and needs help with Person1's business. Person1 has good kids and a good relationship with mom. Person1 hates running. Person2 and Person2 are talking about their plans for high school. Person2 wants to go to Stanford and Person2 wants to go to UC Berkeley. **Based on the given persona details about the Person1 and the Person2, predict what Person2 would say after the Person1 said this dialogue. Person1:** FOR sure! WHAT type of music have you downloaded? WHO is your favorite musician? **Person2:** It's a good idea. I've downloaded a lot of music. I like the Beatles. |
| Persona + Summary Few Shot | **Learn from the example first. Example: Here is some information about Person1 and Person2:** Person2 has been to Boston but grew up in San Francisco. Person2 works a lot and fishs in his spare time. Person2 is married with children. Person1 is an English teacher in Boston. Person1 likes drawing and Fish 'n Chips. Person1 like Batman and Fish 'n Chips. **Here is a conversation summary between Person1 and Person2:** Person2 would love to draw San and volunteer for the homeless. Person1 spends time fishing and traveling. Person2 loves the dark knight movie and often dresses up for cosplay. Person1's wife makes Person1 home-made french fries. Person2's favorite movie is the dark knight. Person1'll watch the next one. **Based on the persona information of the Person1 and the Person2 and their conversation summary so far, anticipate what the Person2's response might be to the Person1's next statement. Person1:** I love it all, but any parts with the joker are awesome. WHAT was your favorite part? **Person2:** SAME, I loved the joker! DID you watch the movie joker with joaquin phoenix? **Now try it yourself. Here are some persona details about the Person1 and the Person2:** Person1 has been to Boston but grew up in San Francisco. Person1 works a lot and fishs in his spare time. Person1 is married with children. Person2 is an English teacher in Boston. Person2 likes drawing and Fish 'n Chips. Person2 and Person2 like Batman and Fish 'n Chips. **Here is a conversation summary between Person1 and Person2.** Person2 spends time fishing when not busy with work. Person1 volunteers for the homeless and walks through parks. Person1 loves the dark knight movie and often dresses up for cosplay. Person2's wife makes Person2 home-made french fries. Person2 is going to watch the next one in the series. **Based on the persona information of the Person1 and the Person2 and their conversation summary so far, anticipate what the Person2's response might be to the Person1's next statement. Person1:** SAME, I loved the joker! DID you watch the movie joker with joaquin phoenix? **Person2:** I did. IT was the darkest movie I've seen and very sad. |
| Persona + Summary Zero Shot | **Here are some persona details about the Person1 and Person2:** Person1's cat just gave birth to a kitten. Person1 lives in the city and doesn't like the country. Person1 likes iced tea and Person1 likes going to the zoo. Person2 is a Senior in High School. Person2 is moving in a week. Person2 likes cats and riding a mountain bike. Person2 speaks Spanish. **Here is a conversation summary between Person1 and Person2:** Person1 is moving into a Spanish community next week. Person2 has just got a new haircut and color from her stylist. Person2 loves the zoo and listening to Spanish people. Person2 will help Person1 unpack at her new place. **Based on the persona information of the Person1 and the Person2 and their conversation summary so far, anticipate what the Person2's response might be to the Person1's next statement. Person1:** YOU're right about that, and thanks so much for offering to help me unpack. I'll definitely have some yummy tea ready! **Person2:** SWEET! LITERALLY. I think this move is going to be a great move for you. HAHA get it? |
| Only Knowledge Zero Shot | **Here is some data on the topics the Person1 and Person2 are discussing about:** Thomas Cruise Mapother IV (born July 3, 1962) is an American actor and producer..He started his career at age 19 in the film Endless Love (1981), before making his breakthrough in the comedy Risky Business (1983).After starring in The Color of Money (1986) and Cocktail (1988), Cruise starred opposite Dustin Hoffman in the Academy Award for Best Picture-winning drama Rain Man..For his role as anti-war activist Ron Kovic in the drama Born on the Fourth of July (1989), Cruise received the Golden Globe The series is co-produced by and stars Tom Cruise, whose character is Ethan Hunt, a special agent of the Impossible Missions Force (IMF).The films follow the missions of the IMF's main field team under the leadership of Hunt ..Some characters, such as Luther Stickell (played by Ving Rhames) and Benji Dunn (played by Simon Pegg) have recurring roles in the films .Wonder Woman is a 2017 American superhero film based on the DC Comics character of the same name ..It is the fourth installment in the DC Extended Universe (DCEU).It is the second live action theatrical film featuring Wonder Woman following her debut in 2016's Batman v Superman: Dawn of Justice . **In the light of the information about the topics being discussed by the Person1 and the Person2, predict what might have been said following this dialogue by the Person1. Person1:** YEAH, I did. THE movie isn't bad either. WELL, nice chatting with you! **Person2:** LEONARD nimoy played a role in that tv show. NICE chatting with you too! |

Table 17: Perplexity Optimized Prompt Examples for T0. Template part is shown in bold. The last output for the Person2 is the generated output using T0. Part 1

| Setting | Prompt |
|---|---|
| Only Knowledge Few Shot | **Learn from the example first. Example: Here is some information on the topics being discussed by Person1 and Person2:** The NFL is one of the four major professional sports leagues in North America ..The NFL has no written rule against female players; women would in fact be allowed if they met the league's eligibility requirements ..New Orleans Saints cheerleaders are forbidden from eating in the same restaurant as any NFL player. The University of Iowa's locker room for visiting football teams is completely painted pink ..The highest score ever in a football game occurred in 1916 when Georgia Tech defeated Cumberland 222-0 ..Former Partiots RB BenJarvus Green-Ellis has never fumbled the football in his NFL career .Animals are multicellular organisms that form the biological kingdom Animalia ..Animals range in length from 8.5 millionths of a metre to 33.6 metres (110 ft) **In the light of the information about the topics being discussed by the Person1 and the Person2, predict what might have been said following this dialogue by the Person1. Person1:** HAVE you ever seen the movie julie & julia with meryl streep? IT is so great and makes me very hungry each time I watch it. **Person2:** I don't think I've seen it but I will add it to my list. ANOTHER good actress is emma watson. I thought she was so good in the harry potter series. **Now try it yourself. Here is some data on the topics the Person1 and Person2 are discussing about:** The Academy Awards are given annually by the Academy of Motion Picture Arts and Sciences ..The ceremony was first broadcast on radio in 1930 and televised for the first time in 1953 ..A total of 3,072 Oscars have been awarded from the inception of the award through the 90th .The Academy Awards are given annually by the Academy of Motion Picture Arts and Sciences ..The ceremony was first broadcast on radio in 1930 and televised for the first time in 1953 ..A total of 3,072 Oscars have been awarded from the inception of the award through the 90th .The Academy Awards are given annually by the Academy of Motion Picture Arts and Sciences ..The ceremony was first broadcast on radio in 1930 and televised for the first time in 1953 ..A total of 3,072 Oscars have been awarded from the inception of the award through the 90th . **In the light of the information about the topics being discussed by the Person1 and the Person2, predict what might have been said following this dialogue by the Person1. Person1:** I have to admit I do like watching the best of the nfc and afc compete against each other! SPEAKING of animals, did you know there is a lawyer in switzerland that represents them in court? **Person2:** The University of Iowa's locker room for visiting football teams is completely painted pink ..The highest score ever in a football game occurred in 1916 when Georgia Tech defeated Cumberland 222-0 ..Former Partiots RB BenJarvus Green-Ellis has never fumbled the football in his NFL career . |
| Knowledge + Summary Few Shot | **Take a look at the following example for guidance. Example: Here is some data on the topics the Person1 and Person2 are discussing about:** William Shakespeare was an English poet, playwright and actor ..He is widely regarded as the greatest writer in the English language ..His plays are performed more often than those of any other playwright .Comedy is a genre of film in which the main emphasis is on humour..Demetri Martin was accepted into Harvard Law, but left out of boredom to pursue a career in comedy..Ryan Stiles dropped out of high school to pursue a career in comedy.Oscar Fingal O'Flahertie Wills Wilde (16 October 1854 2013 30 November 1900) was an Irish poet and playwright..He is best remembered for his epigrams and plays, his novel The Picture of Dorian Gray, and the circumstances of his imprisonment and early death. **Here is a conversation summary between Person1 and Person2:** Person1 and Person2 are football fans. Person2 likes the 49ers because Person2 used to live in San Francisco. Person1 is a hawks fan. Person2 doesn't like brady or belichick. Person1 doesn't know what brady has on facebook. Person2 might not watch him on the facebook watch. **Based on the chat history and the data on topics that the Person1 and the Person2 are chatting about, predict what might have been said following this dialogue by the Person1. Person1:** SO was I but when you stop and think about it, it makes sense. MOST plays only take seconds to complete. THE rest of the time is ads and huddles. **Person2:** YOU are right. I guess 11 minutes of gameplay makes sense. I've got to run now. IT was sincerely nice chatting with you. **Now try it yourself. Here is some data on the topics the Person1 and Person2 are discussing about:** The University of Iowa's locker room for visiting football teams is completely painted pink ..The highest score ever in a football game occurred in 1916 when Georgia Tech defeated Cumberland 222-0 ..Former Partiots RB BenJarvus Green-Ellis has never fumbled the football in his NFL career .The University of Iowa's locker room for visiting football teams is completely painted pink ..The highest score ever in a football game occurred in 1916 when Georgia Tech defeated Cumberland 222-0 ..Former Partiots RB BenJarvus Green-Ellis has never fumbled the football in his NFL career .The University of Iowa's locker room for visiting football teams is completely painted pink ..The highest score ever in a football game occurred in 1916 when Georgia Tech defeated Cumberland 222-0 ..Former Partiots RB BenJarvus Green-Ellis has never fumbled the football in his NFL career . **Here is a conversation summary between Person1 and Person2.** Person2 is a fan of William Shakespeare. Person1 is a fan of Shakespeare's works. Person2 and Person1 both like comedies. Person2 likes old school stuff like monty python. Person1 likes black comedies, including black comedies, and bromances too. **Based on the chat history and the data on topics that the Person1 and the Person2 are chatting about, predict what might have been said following this dialogue by the Person1. Person1:** YEAH, jack black is a natural comedian. I never saw green lantern. ANYWAY, great chat! **Person2:** It's been great talking with you too. I'm a big fan of Monty Python and I think their humor still stands the test of time. |
| Knowledge + Summary Zero Shot | **Here is some data on the topics the Person1 and Person2 are discussing about:** The University of Iowa's locker room for visiting football teams is completely painted pink ..The highest score ever in a football game occurred in 1916 when Georgia Tech defeated Cumberland 222-0 ..Former Partiots RB BenJarvus Green-Ellis has never fumbled the football in his NFL career .The NFL is one of the four major professional sports leagues in North America ..The NFL has no written rule against female players; women would in fact be allowed if they met the league's eligibility requirements ..New Orleans Saints cheerleaders are forbidden from eating in the same restaurant as any NFL player .The Bible is a collection of sacred texts or scriptures that Jews and Christians consider to be a product of divine inspiration and a record of the relationship between God and humans..The Christian Old Testament overlaps with the Hebrew Bible and the Greek Septuagint; the Hebrew Bible is known in Judaism as the Tanakh..The New Testament is a collection of writings by early Christians, believed to be mostly Jewish disciples of Christ, written in first-century Koine Greek. **Here is a conversation summary between Person1 and Person2:** Person2 and Person1 are talking about sports. They talk about Brady's success, the highest scoring football game ever, the 11 minutes of live game play, and the tracking chips on the footballs. **In the light of the chat history and information about topics that the Person1 and the Person2 are chatting on, predict what might have been said following this dialogue by the Person1. Person1:** YES. I think we'll see a woman get in there at some point soon. THERE's no written rule against having female players in the nfl. **Person2:** The NFL has no written rule against female players; women would in fact be allowed if they met the league's eligibility requirements ..New Orleans Saints cheerleaders are forbidden from eating in the same restaurant as any NFL player . |

Table 18: Perplexity Optimized Prompt Examples for T0. Template part is shown in bold. The last output for the Person2 is the generated output using T0. Part 2

| Setting | Prompt |
|---|---|
| Summary Few Shot | **Learn from the example first. Example: Here is a conversation summary between Person1 and Person2:** Person2 prefers to watch movies and cook while Person1 works at a seafood restaurant. Person2's favorite artist is Katy Perry and Person1's favorite band is 21 pilots. They plan to go rock climbing and watch movies later. **In the light of the provided conversation summary between the Person1 and the Person2, predict what might have been said following this dialogue by the Person1. Person1:** THAT one is really good. I like roar. :) **Person2:** OH that's a great one! I also love 21 pilots too. THEIR songs are so catchy. HAVE you ever seen them live? **Now try it yourself. Here is a conversation summary between Person1 and Person2:** Person2 got a call from a place where she put in a job application. Person2 has an interview on Monday. Person1 has been working at the government for 15 years. **In the light of the provided conversation summary between the Person1 and the Person2, predict what might have been said following this dialogue by the Person1. Person1:** WELL not sure I enjoy it lots, its not bad but enjoy is a strong word. JUST think though your searches may already be done,. **Person2:** HAHA yes, enjoy is probably a rare word to use about a job. I really hope I get the role, it'll be nice to settle into the new city knowing I have money coming in soon! |
| Summary Zero Shot | **Here is a summary of the conversation between Person1 and Person2:** Person1 tells Person2 about Person1's favorite band, three dog night, and Person1's favorite song. Person1 is a huge fan of rock or metal. Person1 has seen three dog night once but would love to again. **Based on the given summary of the conversation between the Person1 and the Person2, predict what Person2 would say after the Person1 said this dialogue. Person1:** I really enjoy 'mama told me ', one of their lesser known songs, but I really like it. SOMETIMES those are the best songs. **Person2:** I'll give it a listen when I get a chance. HOW's sao paulo like, I want to visit this summer but I am a bit unsure of what to see or do in the city. |
| Only Persona Few Shot | **Learn from the example first. Example: Here is some information about the Person1 and Person2:** Person2 and Person2 used to work for a government paper pusher. Person2 loves books and has a pet lizard. Person2 worked as a government paper pusher. Person1 works as an assistant at a doctor's office and tells Person1 about the job. Person1 thinks it's a great job.. **Based on the information provided about the Person1 and the Person2, predict what the Person2 might have said in response to the Person1's dialogue. Person1:** HOW is it working in a doctor's office? WORKING as a paper pusher is a little slow but I like it so far. **Person2:** JUST same as other office setting but its all a good job. **Now try it yourself. Here is some information about the Person1 and Person2:** Person1 listens to Bruno Mars, Person1 likes horror movies, pizza and ramen, and Person1 likes rock climbing, zombie movies, and zombie movies. Person2 and Person2 like horror movies, rock climbing, and zombie movies. Person2 broke up with Heather, and Person2 works at a restaurant. **Based on the information provided about the Person1 and the Person2, predict what the Person2 might have said in response to the Person1's dialogue. Person1:** HAVE you painted anything interesting recently? **Person2:** I have. I saw this sun beam come through my skylight the other day and it inspired me to to paint a new piece. IT's like a sunburst. |
| Only Persona Zero Shot | **Here are some persona details about the Person1 and Person2:** Person1 was a veterinarian but quit. Person1's favorite travel destination is Maldives. Person1's band is planning to create music next week. Person2 tells Person2 Person2 might quit Person2's job because Person2 doesn't have time to spend time with family and go to vacations. **Based on the given persona details about the Person1 and the Person2, predict what Person2 would say after the Person1 said this dialogue. Person1:** I remember when nirvana got big! THAT was an exciting time for sure. HAVE you been able to see any acts in concert lately? **Person2:** NO, I have not recently, but I would love to go see some more live music again. WHAT about you? |
| Persona + Summary Few Shot | **Take a look at the following example for guidance. Example: Here is some information about Person1 and Person2:** Person2 tells Person2 Person2 has red hair and wants to quit Person2's job. Person2 likes old-school punk rock and cooking. Person1 and Person1 talk about their favorite food, music, and parents. Person1's not in a committed relationship, while Person1's parents taught Person1 to care for others. **Here is a conversation summary between Person1 and Person2:** Person2 wants to quit Person2's job because Person2 needs to support Person2's family. Person1 picked up all the groceries to make enchiladas for Person2. Person2 loves Person1's family and other food. **In the light of the conversation summary and the information provided about the Person1 and the Person2, predict what might have been said following this dialogue by the Person1. Person1:** OH do you like the strawberry flavor? I also have some other flavor at home, I think I have some lemon and apple flavors left, and I can also use those if you want. **Person2:** HAHA how about all three? IT'll make for a memorable meal. **Now try it yourself. Here is some information about the Person1 and Person2:** Person1 tells Person1 Person1 has red hair and wants to quit Person1's job. Person1 likes old-school punk rock and cooking. Person2 and Person2 talk about their favorite food, music, and parents. Person2's not in a committed relationship, while Person2's parents taught Person2 to care for others. **Here is a conversation summary between Person1 and Person2.** Person1 wants to quit the job because Person1 needs to support Person1's family. Person2 picks up all the groceries Person1 needs to make those enchiladas for Person1. Person2 has used Person2's grandma's secret recipe in it and both of Person2's mom approved of it. Person1 loves Person2's family and will make strawberry jell-o for Person2. **In the light of the conversation summary and the information provided about the Person1 and the Person2, predict what might have been said following this dialogue by the Person1. Person1:** HAHA how about all three? IT'll make for a memorable meal. **Person2:** SURE thing! I'll arrange a giant cakes made from all these jellos, that'll be such a sight to behold! |
| Persona + Summary Zero Shot | **Here are some persona details about the Person1 and Person2:** Daria's name is Daria and she works as a landscaper. Person1's name is Person1. Person1 plays basketball and likes shows about lawyers. John edits videos for a living and Person2 wants to build a keyboard. Person2's popular in town, and John's brother is popular too. **Here is a conversation summary between Person1 and Person2:** Person1's landscaping business is really busy this time of year. Everyone wants their beds cleaned out and mulched. Person2 usually gets motivated to work on his landscaping at this time of year. Person2 has started working on building that keyboard. Person1 would like to have a keyboard with a talk to text feature built into it. **Based on the persona information of the Person1 and the Person2 and their conversation summary so far, anticipate what the Person2's response might be to the Person1's next statement. Person1:** I can never take vacation in the summer, but I'm looking forward to some time off in the fall before the leaves fall. THE beach is calling my name. **Person2:** I can never take vacation in the summer, but I'm looking forward to some time off in the fall before the leaves fall. |

Table 19: Perplexity Optimized Prompt Examples for Tk-Instruct. Template part is shown in bold. The last output for the Person2 is the generated output using Tk-Instruct.

| History Signal | BLEURT | | | | | DEB | | | | | METEOR | | | | |
|---|---|---|---|---|---|---|---|---|---|---|---|---|---|---|---|
| | a=0.5 | a=1 | a=2 | a=5 | a=10 | a=0.5 | a=1 | a=2 | a=5 | a=10 | a=0.5 | a=1 | a=2 | a=5 | a=10 |
| BART-D | 3.21 | 19.79 | 752.7 | 4.16E+07 | 3.38E+15 | 4.92 | 46.56 | 4192.7 | 3.18E+09 | 2.23E+19 | 2.00 | 7.75 | 118.0 | 4.82E+05 | 7.59E+11 |
| Pegasus-CD | 2.94 | 17.76 | 649.6 | 3.22E+07 | 2.26E+15 | 4.65 | 44.58 | 4107.0 | 3.31E+09 | 2.49E+19 | 1.86 | 7.14 | 107.2 | 4.15E+05 | 5.57E+11 |
| Pegasus-DS | 3.20 | 19.72 | 749.8 | 4.14E+07 | 2.26E+15 | 4.91 | 46.49 | 4194.1 | 3.2E+09 | 2.28E+19 | 2.00 | 7.78 | 119.5 | 5.05E+05 | 8.57E+11 |
| Pegasus-DS+BI | 2.01 | 12.23 | 453.6 | 2.34E+07 | 1.72E+15 | 3.13 | 29.65 | 2673.8 | 2.01E+09 | 1.33E+19 | 1.27 | 4.93 | 76.0 | 3.40E+05 | 6.83E+11 |
| Recent-1 | 3.82 | 22.22 | 753.0 | 2.94E+07 | 1.35E+15 | 5.92 | 53.35 | 4351.5 | 2.41E+09 | 9.44E+18 | 2.42 | 8.96 | 123.0 | 3.30E+05 | 1.91E+11 |
| Recent-2 | 3.16 | 19.38 | 727.7 | 3.87E+07 | 2.97E+15 | 4.90 | 46.49 | 4197.5 | 3.15E+09 | 2.07E+19 | 2.02 | 7.92 | 122.6 | 4.81E+05 | 5.57E+11 |
| Recent-4 | 2.48 | 15.18 | 570.1 | 3.04E+07 | 2.35E+15 | 3.86 | 36.77 | 3353.3 | 2.59E+09 | 1.77E+19 | 1.60 | 6.34 | 100.7 | 4.37E+05 | 6.42E+11 |
| Recent-8 | 1.86 | 11.35 | 422.7 | 2.20E+07 | 1.65E+15 | 2.92 | 27.91 | 2563.8 | 2.03E+09 | 1.44E+19 | 1.20 | 4.75 | 75.3 | 3.35E+05 | 5.48E+11 |
| Recent-10 | 1.68 | 10.05 | 359.4 | 1.65E+07 | 9.82E+14 | 2.65 | 25.08 | 2249.8 | 1.67E+09 | 1.1E+19 | 1.07 | 4.08 | 59.5 | 1.90E+05 | 1.44E+11 |
| Semantic-1 | 3.83 | 23.51 | 883.8 | 4.71E+07 | 3.59E+15 | 5.65 | 51.11 | 4208.7 | 2.46E+09 | 1.16E+19 | 2.38 | 9.03 | 131.3 | 4.20E+05 | 3.3E+11 |
| Semantic-2 | 3.25 | 19.86 | 743.8 | 3.93E+07 | 2.96E+15 | 4.84 | 44.19 | 3700.9 | 2.24E+09 | 1.07E+19 | 2.04 | 7.88 | 118.2 | 4.28E+05 | 4.57E+11 |
| Semantic-4 | 2.53 | 15.50 | 580.4 | 3.07E+07 | 2.35E+15 | 4.00 | 38.57 | 3602.3 | 2.99E+09 | 2.29E+19 | 1.64 | 6.52 | 104.0 | 4.58E+05 | 6.85E+11 |
| Semantic-8 | 1.83 | 11.07 | 406.8 | 2.04E+07 | 1.46E+15 | 2.78 | 25.58 | 2178.7 | 1.38E+09 | 6.95E+18 | 1.17 | 4.56 | 70.5 | 3.03E+05 | 5.25E+11 |
| Semantic-10 | 1.61 | 9.72 | 353.8 | 1.74E+07 | 1.22E+15 | 2.46 | 22.62 | 1922.5 | 1.22E+09 | 6.16E+18 | 1.03 | 4.02 | 61.9 | 2.67E+05 | 4.76E+11 |
| Full | 1.18 | 6.97 | 245.5 | 1.11E+07 | 7.27E+14 | 1.89 | 18.03 | 1642.9 | 1.27E+09 | 8.74E+18 | 0.76 | 2.91 | 44.3 | 1.92E+05 | 3.51E+11 |

Table 20: UID versus the Metric-Importance Index $a$ (for MSC Dataset)

| History Signal | BLEURT | | | | | DEB | | | | | METEOR | | | | |
|---|---|---|---|---|---|---|---|---|---|---|---|---|---|---|---|
| | a=0.5 | a=1 | a=2 | a=5 | a=10 | a=0.5 | a=1 | a=2 | a=5 | a=10 | a=0.5 | a=1 | a=2 | a=5 | a=10 |
| BART-D | 4.12 | 24.56 | 873.8 | 3.98E+07 | 2.41E+15 | 6.32 | 57.75 | 4848.1 | 2.96E+09 | 1.45E+19 | 2.45 | 8.67 | 109.6 | 2.31E+05 | 9.2E+10 |
| Pegasus-CD | 3.19 | 18.59 | 632.3 | 2.53E+07 | 1.25E+15 | 5.17 | 48.79 | 4348.5 | 3.1E+09 | 1.81E+19 | 1.95 | 6.96 | 89.2 | 2.00E+05 | 9.06E+10 |
| Pegasus-DS | 4.09 | 24.37 | 864.9 | 3.91E+07 | 2.32E+15 | 6.29 | 57.61 | 4848.0 | 2.98E+09 | 1.46E+19 | 2.44 | 8.64 | 109.3 | 2.32E+05 | 9.37E+10 |
| Pegasus-DS+BI | 1.36 | 7.67 | 243.1 | 7.80E+06 | 2.61E+14 | 2.24 | 20.75 | 1782.1 | 1.15E+09 | 5.94E+18 | 0.84 | 2.94 | 36.9 | 8.92E+04 | 6.46E+10 |
| Recent-1 | 4.10 | 24.17 | 839.9 | 3.58E+07 | 1.95E+15 | 6.49 | 60.36 | 5233.5 | 3.44E+09 | 1.77E+19 | 2.52 | 9.09 | 118.9 | 2.71E+05 | 1.12E+11 |
| Recent-2 | 3.45 | 20.35 | 707.3 | 3.01E+07 | 1.63E+15 | 5.50 | 51.54 | 4533.9 | 3.11E+09 | 1.71E+19 | 2.15 | 7.89 | 106.4 | 2.68E+05 | 1.34E+11 |
| Recent-4 | 2.71 | 15.99 | 558.8 | 2.41E+07 | 1.32E+15 | 4.31 | 40.57 | 3598.5 | 2.53E+09 | 1.45E+19 | 1.70 | 6.31 | 87.4 | 2.40E+05 | 1.42E+11 |
| Recent-8 | 2.02 | 11.92 | 415.3 | 1.77E+07 | 9.46E+14 | 3.23 | 30.56 | 2731.8 | 1.97E+09 | 1.18E+19 | 1.27 | 4.73 | 65.7 | 1.84E+05 | 1.18E+11 |
| Recent-10 | 1.83 | 10.77 | 372.2 | 1.55E+07 | 7.89E+14 | 2.95 | 27.94 | 2505.9 | 1.83E+09 | 1.11E+19 | 1.15 | 4.26 | 58.7 | 1.63E+05 | 1.06E+11 |
| Semantic-1 | 4.09 | 24.02 | 831.3 | 3.48E+07 | 1.82E+15 | 6.46 | 60.06 | 5194.7 | 3.39E+09 | 1.72E+19 | 2.47 | 8.76 | 110.6 | 2.26E+05 | 7.8E+10 |
| Semantic-2 | 3.43 | 20.15 | 696.0 | 2.90E+07 | 1.49E+15 | 5.48 | 51.40 | 4528.7 | 3.12E+09 | 1.71E+19 | 2.11 | 7.64 | 100.4 | 2.34E+05 | 1.04E+11 |
| Semantic-4 | 2.65 | 15.34 | 515.2 | 1.96E+07 | 8.6E+14 | 4.37 | 41.73 | 3811.9 | 2.91E+09 | 1.87E+19 | 1.71 | 6.41 | 90.2 | 2.52E+05 | 1.43E+11 |
| Semantic-8 | 1.97 | 11.57 | 398.0 | 1.63E+07 | 8.06E+14 | 3.19 | 30.20 | 2714.2 | 1.98E+09 | 1.2E+19 | 1.23 | 4.49 | 60.6 | 1.58E+05 | 9.45E+10 |
| Semantic-10 | 1.78 | 10.38 | 353.6 | 1.41E+07 | 6.64E+14 | 2.89 | 27.45 | 2474.3 | 1.82E+09 | 1.12E+19 | 1.11 | 4.03 | 53.9 | 1.38E+05 | 8.36E+10 |
| Full | 1.43 | 8.45 | 296.6 | 1.30E+07 | 7.34E+14 | 2.36 | 23.13 | 2218.7 | 1.97E+09 | 1.63E+19 | 0.91 | 3.44 | 49.5 | 1.56E+05 | 1.25E+11 |

Table 21: UID versus the Metric-Importance Index $a$ (for TC Dataset)