# OpenReview forum: "Frugal Prompting for Dialog Models"
_EMNLP/2023/Conference — EMNLP 2023 Findings_

### Official Review · Reviewer_MCUv · 2023-07-25

**Soundness:** 4

**Excitement:**

3: Ambivalent: It has merits (e.g., it reports state-of-the-art results, the idea is nice), but there are key weaknesses (e.g., it describes incremental work), and it can significantly benefit from another round of revision. However, I won't object to accepting it if my co-reviewers champion it.

**Paper Topic And Main Contributions:**

The paper does a comprehensive study on prompt engineering GPT-3 and other LLMs for open-domain dialogue datasets MSC and TC. The authors wish to focus on frugality when prompting, and to quanitfy the trade-off between model quality and inference cost, introduct Usable Information Density (UID) as a metric to measure it.

They find that:
 - Semantic-1 is the best for UID. Higher k on Recent-k and Semantic-k can improve the core metric but reduce UID.
 - Using summaries has better UID vs. full dialog history.
 - Few-shot accuracy does worse than zero-shot.

**Questions For The Authors:**

1) Can the UID metric be redefined with a scale parameter M^a / L? To quantify the value of the tradeoff between quality and inference cost.

2) PEGASUS-DI + BI has a drastically worse UID vs. PEGASUS-DI in Table 4, on both MSC and TC. Is there a reason why? From what I can tell in Section 4, I don’t think the length of the BI is very long.

**Reasons To Accept:**

This is a comprehensive study on how to prompt LLMs for open-domain dialogue tasks with a focus on frugality. There are a wide variety of parameters that are studied: 0-shot vs 1-shot, manual vs perplexity-chosen prompts, dialog history (full, summarized by BART or PEGASUS, recent-k, semantic-k), and also whether to add summarized background information (BI). To the best of my knowledge, this is the only study to look at prompting methods for open-domain dialogue.

**Reasons To Reject:**

1) I am not sure if I am persuaded by the authors’ design of UID = M / L, where M is some metric, and L is the combined length of inputs and outputs averaged over the dataset. The design of the metric quantifies the tradeoff between model quality and inference cost through a ratio. However, there is an implicit assumption that the desired tradeoff between M and L is linear. It may be that the model quality M can be weighted quadratically more than the sequence length L (e.g. I am a startup with a lot of money to burn and am willing to spend more for a better quality chatbot). In this case, a better metric would be M^2 / L, or in general M^a / L for some a. If there is a scaling parameter involved, then the authors’ conclusions on which models perform best on UID are thrown into question. This paper would benefit from a deeper analysis on how to quantify the tradeoff between M and L, e.g. for a = 0.5, 1, 2, 5, 10…. At high enough a, I would assume that full prompting on GPT-3 would have the best UID.

2) The conclusion presented by the authors (that summarized dialogues or Semantic/Recent-1 is the best approach) is not very impactful. While having this shown empirically is important, I’m not sure what readers can take away from this when doing future dialogue research. It’s very intuitive that conversations can have a lot of redundant information between utterances that can be condensed in order to save on sequence length (e.g. in both DialogSum and SAMSum).

**Reproducibility:**

4: Could mostly reproduce the results, but there may be some variation because of sample variance or minor variations in their interpretation of the protocol or method.

**Reviewer Confidence:**

3: Pretty sure, but there's a chance I missed something. Although I have a good feel for this area in general, I did not carefully check the paper's details, e.g., the math, experimental design, or novelty.

---

> ### Author Rebuttal · Authors · 2023-08-28
>
> Thank you for the thoughtful feedback on our paper. We appreciate you highlighting the comprehensive nature of our study and raising important questions about our proposed UID metric. We are grateful that you took the time to provide such insightful and constructive critiques. We will revise our draft to incorporate the valuable suggestions provided by you.
>
> > **Q1: "However, there is an implicit assumption that the desired tradeoff between M and L is linear. … In this case, a better metric would be M^2 / L, or in general M^a / L for some a. … This paper would benefit from a deeper analysis on how to quantify the tradeoff between M and L, e.g. for a = 0.5, 1, 2, 5, 10…. At high enough a, I would assume that full prompting on GPT-3 would have the best UID."**
>
> > **"Can the UID metric be redefined with a scale parameter M^a / L? To quantify the value of the tradeoff between quality and inference cost."**
>
> Thanks for this very interesting suggestion. We agree with the reviewer that our definition of the UID metric assumes a linear trade-off relation between metric and length. An updated formulation of the UID metric, with M raised to an exponent “a”, can be used to capture the importance assigned by the user on the model performance M (e.g. when inference cost is less of a bottleneck).
>
> $UID(a) = \frac{M^a}{L}$
>
> As suggested, we analyzed the accuracy-length trade-off using different values for the parameter “a” to capture various types of user requirements in terms of allowed expenses for the inference process. These UID values are reported below in Table 1 and Table 2. Based on these experiments, we found some very insightful observations. The average UID values (for zero-shot manual prompts) across all the models are shown in Table 1 and 2 below (**for *better visualization* please check**: [https://anonymous.4open.science/api/repo/Frugal-Prompting-Evaluation/file/results/rebuttal_plots/UID_vs_a_table.png](https://anonymous.4open.science/api/repo/Frugal-Prompting-Evaluation/file/results/rebuttal_plots/UID_vs_a_table.png)). These tables show that for both MSC and TC, for DEB and METEOR, as the value of "a" is increased, summary-based dialog history variants tend to become better in terms of UID while recent-k and semantic-k variants tend to become less impressive. Although, in terms of BLEURT (UID), the ranking is in favour of Semantic-1 or 2 and Recent-1 or 2 throughout the complete range of "a" that we've explored. This might be because BLEURT measures normal sentence semantic similarity but not context-response relevance as measured by DEB.
>
> **Table 1: UID vs. the Metric-Importance Index $a$ (for MSC Dataset)**
>
> |  | History Signal | BLEURT |  |  |  |  | DEB |  |  |  |  | METEOR |  |  |  |  |
> |---|---|---|---|---|---|---|---|---|---|---|---|---|---|---|---|---|
> |  |  | a=0.5 | a=1 | a=2 | a=5 | a=10 | a=0.5 | a=1 | a=2 | a=5 | a=10 | a=0.5 | a=1 | a=2 | a=5 | a=10 |
> | **MSC** | BART-D | 3.21 | 19.79 | 752.7 | 4.16E+07 | 3.37E+15| 4.92 | 46.56 | 4192.7 | 3.17E+9| 2.22E+019 | 2.00 | 7.75 | 118.0 | 4.82E+05 | 7.58E+11|
> |  | Pegasus-CD | 2.94 | 17.76 | 649.6 | 3.22E+07 | 2.25E+15| 4.65 | 44.58 | 4107.0 | 3.31E+9| 2.48E+019 | 1.86 | 7.14 | 107.2 | 4.15E+05 | 5.57E+11|
> |  | Pegasus-DS | 3.20 | 19.72 | 749.8 | 4.14E+07 | 3.36E+15| 4.91 | 46.49 | 4194.1 | 3.20E+9| 2.27E+019 | 2.00 | 7.78 | 119.5 | 5.05E+05 | 8.57E+11|
> |  | Pegasus-DS+BI | 2.01 | 12.23 | 453.6 | 2.34E+07 | 1.71E+15| 3.13 | 29.65 | 2673.8 | 2.00E+9| 1.32E+019 | 1.27 | 4.93 | 76.0 | 3.40E+05 | 6.83E+11|
> |  | Recent-1 | 3.82 | 22.22 | 753.0 | 2.94E+07 | 1.34E+15| 5.92 | 53.35 | 4351.5 | 2.40E+9| 9.44E+018 | 2.42 | 8.96 | 123.0 | 3.30E+05 | 1.91E+11|
> |  | Recent-2 | 3.16 | 19.38 | 727.7 | 3.87E+07 | 2.97E+15| 4.90 | 46.49 | 4197.5 | 3.15E+9| 2.06E+019 | 2.02 | 7.92 | 122.6 | 4.81E+05 | 5.56E+11|
> |  | Recent-4 | 2.48 | 15.18 | 570.1 | 3.04E+07 | 2.35E+15| 3.86 | 36.77 | 3353.3 | 2.59E+9| 1.76E+019 | 1.60 | 6.34 | 100.7 | 4.37E+05 | 6.41E+11|
> |  | Recent-8 | 1.86 | 11.35 | 422.7 | 2.20E+07 | 1.65E+15| 2.92 | 27.91 | 2563.8 | 2.02E+9| 1.44E+019 | 1.20 | 4.75 | 75.3 | 3.35E+05 | 5.47E+11|
> |  | Recent-10 | 1.68 | 10.05 | 359.4 | 1.65E+07 | 9.81E+14| 2.65 | 25.08 | 2249.8 | 1.66E+9| 1.10E+019 | 1.07 | 4.08 | 59.5 | 1.90E+05 | 1.43E+11|
> |  | Semantic-1 | 3.83 | 23.51 | 883.8 | 4.71E+07 | 3.59E+15| 5.65 | 51.11 | 4208.7 | 2.46E+9| 1.15E+019 | 2.38 | 9.03 | 131.3 | 4.20E+05 | 3.29E+11|
> |  | Semantic-2 | 3.25 | 19.86 | 743.8 | 3.93E+07 | 2.96E+15| 4.84 | 44.19 | 3700.9 | 2.24E+9| 1.06E+019 | 2.04 | 7.88 | 118.2 | 4.28E+05 | 4.57E+11|
> |  | Semantic-4 | 2.53 | 15.50 | 580.4 | 3.07E+07 | 2.34E+15| 4.00 | 38.57 | 3602.3 | 2.98E+9| 2.28E+019 | 1.64 | 6.52 | 104.0 | 4.58E+05 | 6.84E+11|
> |  | Semantic-8 | 1.83 | 11.07 | 406.8 | 2.04E+07 | 1.46E+15| 2.78 | 25.58 | 2178.7 | 1.38E+9| 6.94E+018 | 1.17 | 4.56 | 70.5 | 3.03E+05 | 5.24E+11|
> |  | Semantic-10 | 1.61 | 9.72 | 353.8 | 1.74E+07 | 1.21E+15| 2.46 | 22.62 | 1922.5 | 1.21E+9| 6.16E+018 | 1.03 | 4.02 | 61.9 | 2.67E+05 | 4.76E+11|
> |  | Full | 1.18 | 6.97 | 245.5 | 1.11E+07 | 7.27E+14| 1.89 | 18.03 | 1642.9 | 1.27E+9| 8.73E+018 | 0.76 | 2.91 | 44.3 | 1.92E+05 | 3.50E+11|
>
> **Table 2: UID vs. the Metric-Importance Index $a$ (for TC Dataset)**
>
> |  | History Signal | BLEURT |  |  |  |  | DEB |  |  |  |  | METEOR |  |  |  |  |
> |---|---|---|---|---|---|---|---|---|---|---|---|---|---|---|---|---|
> |  |  | a=0.5 | a=1 | a=2 | a=5 | a=10 | a=0.5 | a=1 | a=2 | a=5 | a=10 | a=0.5 | a=1 | a=2 | a=5 | a=10 |
> | **TC** | BART-D | 4.12 | 24.56 | 873.8 | 3.98E+07 | 2.41E+15| 6.32 | 57.75 | 4848.1 | 2.96E+9| 1.44E+019 | 2.45 | 8.67 | 109.6 | 2.31E+05 | 9.19E+10|
> |  | Pegasus-CD | 3.19 | 18.59 | 632.3 | 2.53E+07 | 1.25E+15| 5.17 | 48.79 | 4348.5 | 3.10E+9| 1.80E+019 | 1.95 | 6.96 | 89.2 | 2.00E+05 | 9.06E+10|
> |  | Pegasus-DS | 4.09 | 24.37 | 864.9 | 3.91E+07 | 2.32E+15| 6.29 | 57.61 | 4848.0 | 2.97E+9| 1.45E+019 | 2.44 | 8.64 | 109.3 | 2.32E+05 | 9.37E+10|
> |  | Pegasus-DS+BI | 1.36 | 7.67 | 243.1 | 7.80E+06 | 2.60E+14| 2.24 | 20.75 | 1782.1 | 1.15E+9| 5.94E+018 | 0.84 | 2.94 | 36.9 | 8.92E+04 | 6.46E+10|
> |  | Recent-1 | 4.10 | 24.17 | 839.9 | 3.58E+07 | 1.95E+15| 6.49 | 60.36 | 5233.5 | 3.44E+9| 1.77E+019 | 2.52 | 9.09 | 118.9 | 2.71E+05 | 1.12E+11|
> |  | Recent-2 | 3.45 | 20.35 | 707.3 | 3.01E+07 | 1.63E+15| 5.50 | 51.54 | 4533.9 | 3.11E+9| 1.70E+019 | 2.15 | 7.89 | 106.4 | 2.68E+05 | 1.34E+11|
> |  | Recent-4 | 2.71 | 15.99 | 558.8 | 2.41E+07 | 1.31E+15| 4.31 | 40.57 | 3598.5 | 2.53E+9| 1.45E+019 | 1.70 | 6.31 | 87.4 | 2.40E+05 | 1.41E+11|
> |  | Recent-8 | 2.02 | 11.92 | 415.3 | 1.77E+07 | 9.45E+14| 3.23 | 30.56 | 2731.8 | 1.97E+9| 1.17E+019 | 1.27 | 4.73 | 65.7 | 1.84E+05 | 1.18E+11|
> |  | Recent-10 | 1.83 | 10.77 | 372.2 | 1.55E+07 | 7.88E+14| 2.95 | 27.94 | 2505.9 | 1.82E+9| 1.10E+019 | 1.15 | 4.26 | 58.7 | 1.63E+05 | 1.06E+11|
> |  | Semantic-1 | 4.09 | 24.02 | 831.3 | 3.48E+07 | 1.81E+15| 6.46 | 60.06 | 5194.7 | 3.39E+9| 1.71E+019 | 2.47 | 8.76 | 110.6 | 2.26E+05 | 7.79E+10|
> |  | Semantic-2 | 3.43 | 20.15 | 696.0 | 2.90E+07 | 1.48E+15| 5.48 | 51.40 | 4528.7 | 3.11E+9| 1.71E+019 | 2.11 | 7.64 | 100.4 | 2.34E+05 | 1.03E+11|
> |  | Semantic-4 | 2.65 | 15.34 | 515.2 | 1.96E+07 | 8.60E+14| 4.37 | 41.73 | 3811.9 | 2.90E+9| 1.86E+019 | 1.71 | 6.41 | 90.2 | 2.52E+05 | 1.43E+11|
> |  | Semantic-8 | 1.97 | 11.57 | 398.0 | 1.63E+07 | 8.05E+14| 3.19 | 30.20 | 2714.2 | 1.98E+9| 1.19E+019 | 1.23 | 4.49 | 60.6 | 1.58E+05 | 9.44E+10|
> |  | Semantic-10 | 1.78 | 10.38 | 353.6 | 1.41E+07 | 6.64E+14| 2.89 | 27.45 | 2474.3 | 1.82E+9| 1.12E+019 | 1.11 | 4.03 | 53.9 | 1.38E+05 | 8.35E+10|
> |  | Full | 1.43 | 8.45 | 296.6 | 1.30E+07 | 7.33E+14| 2.36 | 23.13 | 2218.7 | 1.96E+9| 1.62E+019 | 0.91 | 3.44 | 49.5 | 1.56E+05 | 1.25E+11|
>
> To fully understand how the rank of various history signals vary over the value of the metric-importance “a”, we plot the rank-order of all history signal types vs. the value of "a" (increased from 0.5 to 10) as show in the figure [here (https://anonymous.4open.science/api/repo/Frugal-Prompting-Evaluation/file/results/rebuttal_plots/bumpplot_MSC_DEB.png)](https://anonymous.4open.science/api/repo/Frugal-Prompting-Evaluation/file/results/rebuttal_plots/bumpplot_MSC_DEB.png). We have reported the rank-order of the history signal types in **Table 3*** as well. This rank order dynamics helps us clearly understand, as we give more and more importance to the model performance and ignore the cost of inference, how the choices over the history signal change. For example, in terms of the UID (DEB) metric on the MSC dataset, the average trend across models is that Recent-1 and Semantic-1 are the recommended ways to summarize the context information if cost is an important factor to the user. Whereas, if cost is of less importance, then longer dialog summaries such as Pegasus-CD and Semantic-4 are recommended approaches for MSC. Additionally, we observe that some models like Recent-10, Semantic-8, Semantic-10 and Full are always bad choices, while some (BART-D, Pegasus-DS) are quite robust across the whole range of values of "a".
>
> (*P.S. - *We had to share these results in tabular format as there is no provision for adding images (plots/graphs) in Openreview.*)
>
> **Table 3: Trend in Ranks of History Signal Types for Different Values of the Metric-Importance Index $a$ (for MSC dataset, DEB metric)**
>
> | MSC | DEB |  |  |  |  |
> |---|---|---|---|---|---|
> |  | a=0.5 | a=1 | a=2 | a=5 | a=10 |
> | BART-D | 3 | 3 | 5 | 3 | 4 |
> | Pegasus-CD | 7 | 6 | 6 | 1 | 1 |
> | Pegasus-DS | 4 | 4 | 4 | 2 | 3 |
> | Pegasus-DS+BI | 10 | 10 | 10 | 11 | 8 |
> | Recent-1 | 1 | 1 | 1 | 8 | 12 |
> | Recent-2 | 5 | 5 | 3 | 4 | 5 |
> | Recent-4 | 9 | 9 | 9 | 6 | 6 |
> | Recent-8 | 11 | 11 | 11 | 10 | 7 |
> | Recent-10 | 13 | 13 | 12 | 12 | 10 |
> | Semantic-1 | 2 | 2 | 2 | 7 | 9 |
> | Semantic-2 | 6 | 7 | 7 | 9 | 11 |
> | Semantic-4 | 8 | 8 | 8 | 5 | 2 |
> | Semantic-8 | 12 | 12 | 13 | 13 | 14 |
> | Semantic-10 | 14 | 14 | 14 | 15 | 15 |
> | Full | 15 | 15 | 15 | 14 | 13 |
>
> We will add this in-depth analysis of the effect from the chosen value of “a” in the updated draft.
>
> > **Q2: "PEGASUS-DI + BI has a drastically worse UID vs. PEGASUS-DI in Table 4, on both MSC and TC. Is there a reason why? From what I can tell in Section 4, I don’t think the length of the BI is very long."**
>
> Actually, the increase in the length of the prompt due to BI is quite significant for both datasets. The difference in length can be observed in Fig 1. and Fig 4. for few shot and zero shot settings, respectively. The increase in prompt length would be more prominent for the few shot scenario as examples of BI have to be added to the prompt also. For example, BI adds 200 and 500 tokens in average to the prompt for MSC and TC datasets, respectively.
> Now, since the performance of PegasusDS+BI is generally comparable to PegasusDS (Fig. 2 and Fig. 3), the increase in prompt input length (between 1.5-2x) causes a significant drop in the UID values.
>
> > **Q3: "The conclusion presented by the authors ... I’m not sure what readers can take away from this when doing future dialogue research. It’s very intuitive that conversations can have a lot of redundant information between utterances that can be condensed in order to save on sequence length (e.g. in both DialogSum and SAMSum)."**
>
> We do have several important takeaways that are worth considering during practical dialog systems development and future research. Overall, we find that using full dialog history, or Semantic-k/Recent-k with large k are not very useful from a UID perspective. Thus, our results have direct practical implications - lower UID configurations like Semantic-1 can be employed to build real-world dialog systems under budget constraints. (L558). Given the empirical nature of the work, we provide reasons wherever possible.
>
> We present several generally applicable analyses for absolute performance and UID in Sections 5.2 and 5.3 resp.
> General analysis from Section 5.2: (1) GPT3 is best; Tk-Instruct performs the worst -- across all metrics and datasets. (2) Semantic-k performs better than Recent-k -- across all metrics, datasets and models. (3) zero shot results are better than few shot results across metrics, models and datasets. This is such a non-intuitive finding. (4) Perplexity optimized prompts lead to shorter prompt sizes but not better accuracy values, across metrics, models and datasets.
>
> General analysis from Section 5.3 for UID: (1) Surprisingly, manually engineered prompts are better on average (2) using one semantically related utterance is the best way of using dialog history. (3) for both Recent-k and Semantic-k, UID reduces with increase in k. (4) Using summaries of the dialog history provides better UID results than using the full dialog history.
>
> Lastly, as evident from Fig. 5-8 and UIDs in Tab. 5-6 (Appendix), not all models are equally capable of interpreting dialog history in various summarized forms. In terms of UID (DEB) FLAN-T5 has difficulty in understanding Semantic-k and Recent-k summarized dialog histories, but works fine with Pegasus-DS. Whereas T0 is much more performant with all forms of summarized dialog histories in both datasets (L531-535). We apologize if the significance of our findings were not clear in Section 5, and gladly accept this suggestion for improving our paper further.
>
> Once again, we sincerely appreciate your time and effort to provide constructive feedback on our work.

---

### Official Review · Reviewer_6RFn · 2023-08-04

**Soundness:** 3

**Excitement:**

3: Ambivalent: It has merits (e.g., it reports state-of-the-art results, the idea is nice), but there are key weaknesses (e.g., it describes incremental work), and it can significantly benefit from another round of revision. However, I won't object to accepting it if my co-reviewers champion it.

**Paper Topic And Main Contributions:**

Given the recent prominence of LLM, the authors investigate the impact of prompts in dialogue tasks using two dialogue datasets and four models in this paper.
They compare manually engineered prompts and perplexity-based prompts in terms of what kind of prompts should be better. It also examines how dialogue history can be included in the prompts.
Assuming that LLM is used via an API, the authors propose the concept of a frugal prompt because the length of a prompt affects its cost, and they propose a measure called UID to measure it.
From experimental results, the authors show that it is not effective to include many contexts, and that using the most semantically relevant utterances is useful in terms of UID.

**Reasons To Accept:**

- They focus on prompt length from the perspective of practical use, not only from that of performance, and propose a metric that considers it. When using LLM via API, it is necessary to consider the cost. Since the cost of many APIs is determined by the number of tokens, this metric is useful in practical use.
- They investigate what prompts are good from the efficiency perspective and provide valuable guidelines for model design.

**Reasons To Reject:**

- In section 5, they just state the experiment's results without much consideration of the reasons for such results. Therefore, the analysis lacks generality and may or may not be useful.
- Their argument needs to have verification of the usefulness of the UID. The UID can be interpreted as the average of the performance values normalized by the length of the prompt, but since the scale is different, using the values as they are, the length effect is likely to be significant.

**Reproducibility:**

4: Could mostly reproduce the results, but there may be some variation because of sample variance or minor variations in their interpretation of the protocol or method.

**Reviewer Confidence:**

2: Willing to defend my evaluation, but it is fairly likely that I missed some details, didn't understand some central points, or can't be sure about the novelty of the work.

---

> ### Author Rebuttal · Authors · 2023-08-28
>
> We thank the reviewer for their insightful and constructive comments on our paper. Their feedback has helped us better contextualize and articulate the contributions of our work. We are grateful for your perspective on the practical usefulness of our work in providing guidance for efficient model design when using LLMs via API. We acknowledge your point on needing more clarity about takeaways and impacts of our findings in the NLP community, and will revise our draft to incorporate your suggestions.
>
> > **Q1: "In section 5, they just state the experiment's results without much consideration of the reasons for such results. Therefore, the analysis lacks generality and may or may not be useful."**
>
> We do have several important takeaways that are worth considering during practical dialog systems development and future research. Reviewers RHz2 and MCUv also point out the comprehensiveness of the experimental findings and lack of a similar study in literature looking at the important matter of optimizing inference costs while developing dialog systems. Overall, we find that using full dialog history, or Semantic-k/Recent-k with large k are not very useful from a UID perspective. Thus, our results have direct practical implications - lower UID configurations like Semantic-1 can be employed to build real-world dialog systems under budget constraints. (L558). Given the empirical nature of the work, we provide reasons wherever possible.
>
> We present several generally applicable analyses for absolute performance and UID in Sections 5.2 and 5.3 resp.
> General analysis from Section 5.2: (1) GPT3 is best; Tk-Instruct performs the worst -- across all metrics and datasets. (2) Semantic-k performs better than Recent-k -- across all metrics, datasets and models. (3) zero shot results are better than few shot results across metrics, models and datasets. This is such a non-intuitive finding. (4) Perplexity optimized prompts lead to shorter prompt sizes but not better accuracy values, across metrics, models and datasets.
>
> General analysis from Section 5.3 for UID: (1) Surprisingly, manually engineered prompts are better on average (2) using one semantically related utterance is the best way of using dialog history. (3) for both Recent-k and Semantic-k, UID reduces with increase in k. (4) Using summaries of the dialog history provides better UID results than using the full dialog history.
>
> Lastly, as evident from Fig. 5-8 and UIDs in Tab. 5-6 (Appendix), not all models are equally capable of interpreting dialog history in various summarized forms. In terms of UID (DEB) FLAN-T5 has difficulty in understanding Semantic-k and Recent-k summarized dialog histories, but works fine with Pegasus-DS. Whereas T0 is much more performant with all forms of summarized dialog histories in both datasets (L531-535). We apologize if the significance of our findings were not clear in Section 5, and gladly accept this suggestion for improving our paper further.
>
> > **Q2: "Their argument needs to have verification of the usefulness of the UID. The UID can be interpreted as the average of the performance values normalized by the length of the prompt, but since the scale is different, using the values as they are, the length effect is likely to be significant."**
>
> We appreciate the reviewer's concern regarding verification of the usefulness of the UID metric. To address this, we have conducted additional experiments analyzing the effect of using different exponents "a" in the following UID formulation
>
> $$UID(a)= \frac{M^a}{L}$$
>
> for different values of $a$ (metric-importance index). We vary a as [0.5, 1, 2, 5, 10]. This updated formulation of the UID metric, with M raised to an exponent “a”, can be used to capture the importance assigned by the user on the model performance M (e.g. when inference cost is less of a bottleneck). We analyzed the accuracy-length tradeoff using different values for the parameter “a” to capture various types of user requirements in terms of the allowed expenses towards the inference process. Based on these experiments, we found some very insightful observations. The average UID values (for zeroshot manual prompts) across all the models are shown in Table 1 and 2 below (**for *better visualization* please check**: [https://anonymous.4open.science/api/repo/Frugal-Prompting-Evaluation/file/results/rebuttal_plots/UID_vs_a_table.png](https://anonymous.4open.science/api/repo/Frugal-Prompting-Evaluation/file/results/rebuttal_plots/UID_vs_a_table.png)). These tables show that for both MSC and TC, for DEB and METEOR, as the value of "a" is increased, summary-based dialog history variants tend to become better in terms of UID while recent-k and semantic-k variants tend to become less impressive. Although, in terms of BLEURT (UID), the ranking is in favour of Semantic-1 or 2 and Recent-1 or 2 throughout the complete range of "a" that we've explored. This might be because BLEURT measures normal sentence semantic similarity but not context-response relevance as measured by DEB.
>
> **Table 1: UID vs. the Metric-Importance Index $a$ (for MSC Dataset)**
>
> |  | History Signal | BLEURT |  |  |  |  | DEB |  |  |  |  | METEOR |  |  |  |  |
> |---|---|---|---|---|---|---|---|---|---|---|---|---|---|---|---|---|
> |  |  | a=0.5 | a=1 | a=2 | a=5 | a=10 | a=0.5 | a=1 | a=2 | a=5 | a=10 | a=0.5 | a=1 | a=2 | a=5 | a=10 |
> | **MSC** | BART-D | 3.21 | 19.79 | 752.7 | 4.16E+07 | 3.37E+15| 4.92 | 46.56 | 4192.7 | 3.17E+9| 2.22E+019 | 2.00 | 7.75 | 118.0 | 4.82E+05 | 7.58E+11|
> |  | Pegasus-CD | 2.94 | 17.76 | 649.6 | 3.22E+07 | 2.25E+15| 4.65 | 44.58 | 4107.0 | 3.31E+9| 2.48E+019 | 1.86 | 7.14 | 107.2 | 4.15E+05 | 5.57E+11|
> |  | Pegasus-DS | 3.20 | 19.72 | 749.8 | 4.14E+07 | 3.36E+15| 4.91 | 46.49 | 4194.1 | 3.20E+9| 2.27E+019 | 2.00 | 7.78 | 119.5 | 5.05E+05 | 8.57E+11|
> |  | Pegasus-DS+BI | 2.01 | 12.23 | 453.6 | 2.34E+07 | 1.71E+15| 3.13 | 29.65 | 2673.8 | 2.00E+9| 1.32E+019 | 1.27 | 4.93 | 76.0 | 3.40E+05 | 6.83E+11|
> |  | Recent-1 | 3.82 | 22.22 | 753.0 | 2.94E+07 | 1.34E+15| 5.92 | 53.35 | 4351.5 | 2.40E+9| 9.44E+018 | 2.42 | 8.96 | 123.0 | 3.30E+05 | 1.91E+11|
> |  | Recent-2 | 3.16 | 19.38 | 727.7 | 3.87E+07 | 2.97E+15| 4.90 | 46.49 | 4197.5 | 3.15E+9| 2.06E+019 | 2.02 | 7.92 | 122.6 | 4.81E+05 | 5.56E+11|
> |  | Recent-4 | 2.48 | 15.18 | 570.1 | 3.04E+07 | 2.35E+15| 3.86 | 36.77 | 3353.3 | 2.59E+9| 1.76E+019 | 1.60 | 6.34 | 100.7 | 4.37E+05 | 6.41E+11|
> |  | Recent-8 | 1.86 | 11.35 | 422.7 | 2.20E+07 | 1.65E+15| 2.92 | 27.91 | 2563.8 | 2.02E+9| 1.44E+019 | 1.20 | 4.75 | 75.3 | 3.35E+05 | 5.47E+11|
> |  | Recent-10 | 1.68 | 10.05 | 359.4 | 1.65E+07 | 9.81E+14| 2.65 | 25.08 | 2249.8 | 1.66E+9| 1.10E+019 | 1.07 | 4.08 | 59.5 | 1.90E+05 | 1.43E+11|
> |  | Semantic-1 | 3.83 | 23.51 | 883.8 | 4.71E+07 | 3.59E+15| 5.65 | 51.11 | 4208.7 | 2.46E+9| 1.15E+019 | 2.38 | 9.03 | 131.3 | 4.20E+05 | 3.29E+11|
> |  | Semantic-2 | 3.25 | 19.86 | 743.8 | 3.93E+07 | 2.96E+15| 4.84 | 44.19 | 3700.9 | 2.24E+9| 1.06E+019 | 2.04 | 7.88 | 118.2 | 4.28E+05 | 4.57E+11|
> |  | Semantic-4 | 2.53 | 15.50 | 580.4 | 3.07E+07 | 2.34E+15| 4.00 | 38.57 | 3602.3 | 2.98E+9| 2.28E+019 | 1.64 | 6.52 | 104.0 | 4.58E+05 | 6.84E+11|
> |  | Semantic-8 | 1.83 | 11.07 | 406.8 | 2.04E+07 | 1.46E+15| 2.78 | 25.58 | 2178.7 | 1.38E+9| 6.94E+018 | 1.17 | 4.56 | 70.5 | 3.03E+05 | 5.24E+11|
> |  | Semantic-10 | 1.61 | 9.72 | 353.8 | 1.74E+07 | 1.21E+15| 2.46 | 22.62 | 1922.5 | 1.21E+9| 6.16E+018 | 1.03 | 4.02 | 61.9 | 2.67E+05 | 4.76E+11|
> |  | Full | 1.18 | 6.97 | 245.5 | 1.11E+07 | 7.27E+14| 1.89 | 18.03 | 1642.9 | 1.27E+9| 8.73E+018 | 0.76 | 2.91 | 44.3 | 1.92E+05 | 3.50E+11|
>
> **Table 2: UID vs. the Metric-Importance Index $a$ (for TC Dataset)**
>
> |  | History Signal | BLEURT |  |  |  |  | DEB |  |  |  |  | METEOR |  |  |  |  |
> |---|---|---|---|---|---|---|---|---|---|---|---|---|---|---|---|---|
> |  |  | a=0.5 | a=1 | a=2 | a=5 | a=10 | a=0.5 | a=1 | a=2 | a=5 | a=10 | a=0.5 | a=1 | a=2 | a=5 | a=10 |
> | **TC** | BART-D | 4.12 | 24.56 | 873.8 | 3.98E+07 | 2.41E+15| 6.32 | 57.75 | 4848.1 | 2.96E+9| 1.44E+019 | 2.45 | 8.67 | 109.6 | 2.31E+05 | 9.19E+10|
> |  | Pegasus-CD | 3.19 | 18.59 | 632.3 | 2.53E+07 | 1.25E+15| 5.17 | 48.79 | 4348.5 | 3.10E+9| 1.80E+019 | 1.95 | 6.96 | 89.2 | 2.00E+05 | 9.06E+10|
> |  | Pegasus-DS | 4.09 | 24.37 | 864.9 | 3.91E+07 | 2.32E+15| 6.29 | 57.61 | 4848.0 | 2.97E+9| 1.45E+019 | 2.44 | 8.64 | 109.3 | 2.32E+05 | 9.37E+10|
> |  | Pegasus-DS+BI | 1.36 | 7.67 | 243.1 | 7.80E+06 | 2.60E+14| 2.24 | 20.75 | 1782.1 | 1.15E+9| 5.94E+018 | 0.84 | 2.94 | 36.9 | 8.92E+04 | 6.46E+10|
> |  | Recent-1 | 4.10 | 24.17 | 839.9 | 3.58E+07 | 1.95E+15| 6.49 | 60.36 | 5233.5 | 3.44E+9| 1.77E+019 | 2.52 | 9.09 | 118.9 | 2.71E+05 | 1.12E+11|
> |  | Recent-2 | 3.45 | 20.35 | 707.3 | 3.01E+07 | 1.63E+15| 5.50 | 51.54 | 4533.9 | 3.11E+9| 1.70E+019 | 2.15 | 7.89 | 106.4 | 2.68E+05 | 1.34E+11|
> |  | Recent-4 | 2.71 | 15.99 | 558.8 | 2.41E+07 | 1.31E+15| 4.31 | 40.57 | 3598.5 | 2.53E+9| 1.45E+019 | 1.70 | 6.31 | 87.4 | 2.40E+05 | 1.41E+11|
> |  | Recent-8 | 2.02 | 11.92 | 415.3 | 1.77E+07 | 9.45E+14| 3.23 | 30.56 | 2731.8 | 1.97E+9| 1.17E+019 | 1.27 | 4.73 | 65.7 | 1.84E+05 | 1.18E+11|
> |  | Recent-10 | 1.83 | 10.77 | 372.2 | 1.55E+07 | 7.88E+14| 2.95 | 27.94 | 2505.9 | 1.82E+9| 1.10E+019 | 1.15 | 4.26 | 58.7 | 1.63E+05 | 1.06E+11|
> |  | Semantic-1 | 4.09 | 24.02 | 831.3 | 3.48E+07 | 1.81E+15| 6.46 | 60.06 | 5194.7 | 3.39E+9| 1.71E+019 | 2.47 | 8.76 | 110.6 | 2.26E+05 | 7.79E+10|
> |  | Semantic-2 | 3.43 | 20.15 | 696.0 | 2.90E+07 | 1.48E+15| 5.48 | 51.40 | 4528.7 | 3.11E+9| 1.71E+019 | 2.11 | 7.64 | 100.4 | 2.34E+05 | 1.03E+11|
> |  | Semantic-4 | 2.65 | 15.34 | 515.2 | 1.96E+07 | 8.60E+14| 4.37 | 41.73 | 3811.9 | 2.90E+9| 1.86E+019 | 1.71 | 6.41 | 90.2 | 2.52E+05 | 1.43E+11|
> |  | Semantic-8 | 1.97 | 11.57 | 398.0 | 1.63E+07 | 8.05E+14| 3.19 | 30.20 | 2714.2 | 1.98E+9| 1.19E+019 | 1.23 | 4.49 | 60.6 | 1.58E+05 | 9.44E+10|
> |  | Semantic-10 | 1.78 | 10.38 | 353.6 | 1.41E+07 | 6.64E+14| 2.89 | 27.45 | 2474.3 | 1.82E+9| 1.12E+019 | 1.11 | 4.03 | 53.9 | 1.38E+05 | 8.35E+10|
> |  | Full | 1.43 | 8.45 | 296.6 | 1.30E+07 | 7.33E+14| 2.36 | 23.13 | 2218.7 | 1.96E+9| 1.62E+019 | 0.91 | 3.44 | 49.5 | 1.56E+05 | 1.25E+11|
>
>
> To fully understand how the rank of various history signals vary over the value of the metric-importance “a”, we plot the rank-order of all history signal types vs. the value of "a" (increased from 0.5 to 10) as show in the figure [here (https://anonymous.4open.science/api/repo/Frugal-Prompting-Evaluation/file/results/rebuttal_plots/bumpplot_MSC_DEB.png)](https://anonymous.4open.science/api/repo/Frugal-Prompting-Evaluation/file/results/rebuttal_plots/bumpplot_MSC_DEB.png). We have reported the rank-order of the history signal types in **Table 3*** as well. This rank order dynamics helps us clearly understand, as we give more and more importance to the model performance and ignore the cost of inference, how the choices over the history signal change. For example, in terms of the UID (DEB) metric on the MSC dataset, the average trend across models is that Recent-1 and Semantic-1 are the recommended ways to summarize the context information if cost is an important factor to the user. Whereas, if cost is of less importance, then longer dialog summaries such as Pegasus-CD and Semantic-4 are recommended approaches for MSC. Additionally, we observe that some models like Recent-10, Semantic-8, Semantic-10 and Full are always bad choices, while some (BART-D, Pegasus-DS) are quite robust across the whole range of values of "a".
>
> (*P.S. - *We had to share these results in tabular format as there is no provision for adding images (plots/graphs) in Openreview.*)
>
> **Table 3: Trend in Ranks of History Signal Types for Different Values of the Metric-Importance Index $a$ (for MSC dataset, DEB metric)**
>
> | MSC | DEB |  |  |  |  |
> |---|---|---|---|---|---|
> |  | a=0.5 | a=1 | a=2 | a=5 | a=10 |
> | BART-D | 3 | 3 | 5 | 3 | 4 |
> | Pegasus-CD | 7 | 6 | 6 | 1 | 1 |
> | Pegasus-DS | 4 | 4 | 4 | 2 | 3 |
> | Pegasus-DS+BI | 10 | 10 | 10 | 11 | 8 |
> | Recent-1 | 1 | 1 | 1 | 8 | 12 |
> | Recent-2 | 5 | 5 | 3 | 4 | 5 |
> | Recent-4 | 9 | 9 | 9 | 6 | 6 |
> | Recent-8 | 11 | 11 | 11 | 10 | 7 |
> | Recent-10 | 13 | 13 | 12 | 12 | 10 |
> | Semantic-1 | 2 | 2 | 2 | 7 | 9 |
> | Semantic-2 | 6 | 7 | 7 | 9 | 11 |
> | Semantic-4 | 8 | 8 | 8 | 5 | 2 |
> | Semantic-8 | 12 | 12 | 13 | 13 | 14 |
> | Semantic-10 | 14 | 14 | 14 | 15 | 15 |
> | Full | 15 | 15 | 15 | 14 | 13 |
>
> We will add this in-depth analysis of the effect from the chosen value of “a” in the updated draft. Once again, we sincerely appreciate your time and effort to provide constructive feedback on our work.

---

### Official Review · Reviewer_RHz2 · 2023-08-04

**Typos Grammar Style And Presentation Improvements:** (573-574) It is important to clarify …
**Soundness:** 4

**Excitement:**

4: Strong: This paper deepens the understanding of some phenomenon or lowers the barriers to an existing research direction.

**Paper Topic And Main Contributions:**

This paper addresses the problem of prompt optimization in the context of large language models. More specifically, it explores the trade-off between prompt length and performance when such models are used as dialog systems. The paper describes extensive experiments that compare multiple approaches to reduce prompt length while maintaining sufficient performance. The experiments mainly focus on different ways to compress the dialog history and background information, as well as on the comparison of zero-shot and few-shot scenarios. The results in terms of usable information density, a new metric defined by the authors, on two datasets using four different large language models suggest that the highest efficiency can be achieved using zero-shot scenarios and by reducing the dialog history to the most semantically similar utterance.

**Reasons To Accept:**

The paper describes extensive experiments on prompt optimization for the trade-off between length and performance. This is a relevant problem, considering that the costs increase with length, especially when using the paid APIs of many recent large language models.

**Reasons To Reject:**

Considering the extent of the experiments, as revealed by the length of the supplementary material, it could be more appropriate to report them in a journal article.

**Reproducibility:**

4: Could mostly reproduce the results, but there may be some variation because of sample variance or minor variations in their interpretation of the protocol or method.

**Reviewer Confidence:**

4: Quite sure. I tried to check the important points carefully. It's unlikely, though conceivable, that I missed something that should affect my ratings.

---

> ### Author Rebuttal · Authors · 2023-08-28
>
> We deeply appreciate the reviewer providing thoughtful and constructive feedback on our work. Your positive outlook about our work encourages us. We are pleased that you found our extensive experiments on prompt optimization impactful for understanding the trade-off between length and performance. Given the widespread use of large language models across domains, we are delighted that you consider the research question addressed in this paper to be relevant and meaningful.
>
> > **Q1. "(573-574) It is important to clarify what efficiency means in this context."**
>
>
> Thanks for identifying this lack of clarity in usage of the term “efficiency” in the draft.
> In this work, by the term “efficiency” in L573-574 and all other occurrences, we refer to the temporal efficiency of the inference step of transformer-based language generation models. The decoding time during inference depends on the length of the input prompt and model scale. For a model of given scale, the more the length of the input, the more the compute requirements and hence the increased time and API cost. We will clarify this in the revised version.

---

### Meta-Review · Area_Chair_ezmV · 2023-09-12

**Recommendation:** 3

**Metareview:**

The paper explores prompt optimization for dialogue applications (focusing on open-domain chit chat).

**Pros**: All reviewers agree the study is fairly comprehensive, and most mention that the problem is important/relevant. There are limited technical concerns overall, and most concerns raised appear to be resolved during the rebuttal.

**Cons**: There is some disagreement among reviewers on the impact of the work. While reviewers all agree on the comprehensiveness of the study, some reviewers also feel there are issues in interpretation of the results, indicating a lack of generalizable takeaways and concerns on the proposed UID metric. Authors provide a detailed rebuttal with new results and highlight relevant takeaways from the current iteration - both of which are acknowledged by reviewers - but excitement is still marginal.

---

### Decision · Program_Chairs · 2023-10-07

**Decision:**

Accept-Findings

**Comment:**

The paper explores prompt optimization for dialogue applications (focusing on open-domain chit chat).

**Pros**: All reviewers agree the study is fairly comprehensive, and most mention that the problem is important/relevant. There are limited technical concerns overall, and most concerns raised appear to be resolved during the rebuttal.

**Cons**: There is some disagreement among reviewers on the impact of the work. While reviewers all agree on the comprehensiveness of the study, some reviewers also feel there are issues in interpretation of the results, indicating a lack of generalizable takeaways and concerns on the proposed UID metric. Authors provide a detailed rebuttal with new results and highlight relevant takeaways from the current iteration - both of which are acknowledged by reviewers - but excitement is still marginal.